# FEDERATED GRAPH-LEVEL CLUSTERING NETWORK WITH DUAL KNOWLEDGE SEPARATION

**Xiaobao Wang[1], Renda Han[1], Ronghao Fu[2], Di Jin[3,*]**

[1] School of Artificial Intelligence, Tianjin University, Tianjin, 300000, China

[2] College of Computer Science and Technology, Jilin University, Changchun, 130000, China

[3] School of Computer Science and Technology, Tianjin University, Tianjin, 300000, China

{wangxiaobao, jindi}@tju.edu.cn, rd.h@computer.org, furh@jlu.edu.cn

## ABSTRACT

Federated Graph-level Clustering (FGC) offers a promising framework for analyzing distributed graph data while ensuring privacy protection. However, existing methods fail to simultaneously consider knowledge heterogeneity across intra- and inter-client, and still attempt to share as much knowledge as possible, resulting in consensus failure in the server. To solve these issues, we propose a novel **F**ederated **G**raph-level **C**lustering Network with **D**ual **K**nowledge **S**eparation (FGCN-DKS). The core idea is to decouple differentiated subgraph patterns and optimize them separately on the client, and then leverage cluster-oriented patterns to guide personalized knowledge aggregation on the server. Specifically, on the client, we separate personalized subgraphs and cluster-oriented subgraphs for each graph. Then the former are retained locally for further refinement of the clustering process, while pattern digests are extracted from the latter for uploading to the server. On the server, we calculate the relation of inter-cluster patterns to adaptively aggregate cluster-oriented prototypes and parameters. Finally, the server generates personalized guidance signals for each cluster of clients, which are then fed back to local clients to enhance overall clustering performance. Extensive experiments on multiple graph benchmark datasets have proven the superiority of the proposed FGCN-DKS over the SOTA methods.

## 1 INTRODUCTION

Federated Graph Learning (FGL) (Liang et al., 2023; 2024b; Liu et al., 2024a;b; Li & Guo, 2025) has recently emerged as a powerful paradigm for privacy-preserving machine learning (Liao et al., 2026; 2025b;a), enabling multiple clients to collaboratively train models without exposing their raw graph data. With the explosive growth of graph-structured data in domains such as personalized recommendation (Wu et al., 2021), decentralized fraud detection (Chen et al., 2024), and scientific discovery (Zhang et al., 2023; Liang et al., 2024c), research on FGL has gained increasing attention.

Among the various tasks in this domain, clustering (Zhang et al., 2024; Bo et al., 2020) plays a fundamental role by discovering latent patterns without label supervision. In federated settings, clustering can be performed at different granularities, which leads to two distinct paradigms: *node-level* and *graph-level* clustering. In federated node-level clustering (Liang et al., 2024a; Liu et al., 2023; 2025b), clients hold subgraphs drawn from the same global graph, where distributions are relatively homogeneous, allowing the server to easily achieve consensus. In contrast, federated graph-level clustering (FGC) (Liang et al., 2024c)

| Node-Level | | Graph-Level | | |
|---|---|---|---|---|
| **Datasets** | **hr$_O$** | **NS** | **hr$_O$** | **hr$_I$** |
| Cite | 23.7 | SM | 45.2 | 54.5 |
| PubMed | 18.6 | SM-BIO | 69.1 | 58.3 |
| Photo | 4.4 | SN | 43.6 | 39.6 |

Table 1: Multi-subgraph/graph heterogeneity in node- and graph-level tasks, calculated by graph kernel. Here, hr$_O$, hr$_I$ denote inter- and intra-client heterogeneity, respectively. NS refers to non-IID settings (i.e., the strategy of assigning different private datasets to clients).

---

*Corresponding author

requires clients to cluster entirely different non-IID graphs. This introduces severe **intra-client** heterogeneity (inconsistent graph patterns within each client) and **inter-client** heterogeneity (domain shifts across clients), making server consensus much more challenging (see Table 1). Recent methods such as FedGCN (Liu et al., 2025a) and FedPKA (Wu et al., 2025) follow the paradigm of maximizing global knowledge sharing that works for graph-level tasks, but they overlook the unique challenges of multi-graph heterogeneity. As a result, they often suffer from consensus failure when applied to graph-level clustering.

Inspired by FedPer (Arivazhagan et al., 2019), which separates model parameters across layers to enable personalized training in FL, we extend this idea to graph structure by exploring how graphs can be decomposed into different components for FGC. Guided by Invariant Graph Learning (IGL) (Sui et al., 2024; Li et al., 2022b), we further attempt to separate each graph into *common* and *personalized* subgraphs: common parts are shared with the server to support global consensus, while personalized parts are kept locally to protect unique knowledge, as shown in Fig. 1 (a). This design directly matches our goal: common subgraphs contain stable cross-domain patterns (see Fig. 1 (b)), whereas variant subgraphs represent client-oriented information. However, deploying IGL in federated settings is highly challenging. On each client, multiple graphs with diverse distributions coexist, so the extraction of common components must be carefully controlled in granularity to benefit both local clustering and global consensus. On the server, client heterogeneity prevents simple weight aggregation. The global model must move beyond naive consensus and accurately identify representative patterns across participants, guiding a more optimal aggregation process.

Based on the above challenges, we propose **F**ederated **G**raph-Level **C**lustering **N**etwork with **D**ual **K**nowledge **S**eparation (FGCN-DKS). The key idea is to decouple graph knowledge within and across clients, so that local clustering benefits from personalization while the server achieves consensus. On each client, an invariant subgraph separator divides graphs into cluster-oriented common subgraphs and client-specific personalized subgraphs. Only common knowledge digests are uploaded to the server, while personalized subgraphs remain local. On the server, a Common Knowledge Sharing Strategy (CKSS) aggregates invariant pattern digests by computing cluster-level affinities and capturing semantically consistent components. The resulting cluster-level signals are sent back to clients, enabling finer-grained consensus. Finally, clustering is conducted in two stages: $K$-means is initialized with common representations that are extracted from common parts and refined with personalized representations for local adaptation. In summary, our contributions are threefold:

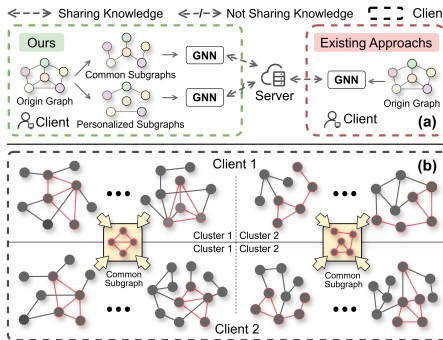

Figure 1: (a) Compared with the existing FGL methods, our approach only shares knowledge that is beneficial to global consensus. (b) Graphs within the same cluster share certain common substructures, and there is also inter-cluster sharing across different clients.

- **New Perspective.** We provide the first systematic study of federated graph-level clustering (FGC) under both intra-client and inter-client heterogeneity, revealing why existing paradigms of maximizing global knowledge sharing fail in this more challenging setting.

- **New Method.** We propose FGCN-DKS, a dual knowledge separation framework that separates invariant and variant subgraphs on clients and performs cluster-level consensus aggregation on the server, directly addressing the identified challenges.

- **Strong Results.** Extensive experiments demonstrate that FGCN-DKS consistently outperforms state-of-the-art baselines in graph clustering performance.

## 2 RELATED WORK

### 2.1 INVARIANT GRAPH LEARNING

Learning graph representations that remain stable under distributional shifts has become a central theme in out-of-distribution (OOD) generalization. Early work, such as GIL (Li et al., 2022a),

introduces a subgraph generator and invariant learning module to extract label substructures, inferring latent environments via variant subgraphs and enforcing consistency across them. Building on this, CIGA (Chen et al., 2022) employs an information-theoretic objective to identify subgraphs whose embeddings maximize intra-class invariance under diverse graph interventions. At the cluster level, CIT (Xia et al., 2023) ensures that cluster embeddings remain consistent despite structural perturbations, promoting robust GNN representations. Beyond task-specific frameworks, several general-purpose techniques further the cause of invariance in graphs. MARIO (Zhu et al., 2024) integrates an Information bottleneck with adversarial augmentations in graph contrastive learning to distill invariant features. CGCL (Chen et al., 2025) enforces cross-view reconstruction consistency between augmented graph views, enhancing OOD robustness for link prediction. IGM (Jia et al., 2024) synthesizes new environments via env-Mixup and inv-Mixup on variant and invariant subgraphs, obviating the need for manual environment labels. More recent advances continue to push the task. InfoIGL (Mao et al., 2024) leverages a multi-level contrastive learning grounded in the Information Bottleneck principle to isolate invariant graph features. MPHIL (Shen et al., 2025) introduces hyperspherical invariant representations with multi-prototype matching and separation losses, directly tackling semantic entanglement across unknown environments. Despite these advances, existing methods assume centralized access to fully labeled data, which is incompatible with federated settings where clients neither share raw data nor possess label supervision.

## 2.2 FEDERATED GRAPH LEARNING

FGL has emerged to enable collaborative model training across multiple clients while preserving data privacy. A straightforward extension of FedAvg (Li et al., 2019) to GNN demonstrates that naively averaging GNN parameters can yield reasonable performance but suffers under the non-IID issue. Subsequently, FedPer (Arivazhagan et al., 2019) adapts personalization layers in GNNs, enabling clients to fine-tune private parameters while sharing a common backbone. FedProx (Li et al., 2020) generalizes and re-parametrizes FedAvg, providing convergence guarantees when learning from non-IID datasets. To address heterogeneity, FedGraphNN (He et al., 2021) introduces client-specific adaptation layers and a global graph aggregator, improving convergence in graph classification and node prediction tasks. Building on this, FedSage (Zhang et al., 2021) and FedGAT (Ambekar et al., 2024) incorporate sampling-based neighbor selection and attention mechanisms, respectively, to reduce communication overhead and align local and global feature spaces. In parallel, FedStar (Tan et al., 2023) addresses client label heterogeneity by aligning local embeddings via contrastive regularization. More recent research, FedGCN (Liu et al., 2025a), as a first FGC framework, is proposed, which optimizes prototypes between multiple clients and guides the local model to learn. Subsequently, FedPKA (Wu et al., 2025) mitigates non-IID heterogeneity and knowledge drift by confidence-guided knowledge aggregation and adaptive prototype adjustment for personalized FL. However, existing methods still cannot effectively solve the consensus failure issue caused by large knowledge differences. In contrast, FGCN-DKS effectively alleviates it through client internal knowledge decoupling and cluster-oriented personalized aggregation between clients.

## 3 METHODOLOGY

In this section, we present the proposed Federated Graph Learning framework called **F**ederated **G**raph **C**lustering **N**etwork with **D**ual **S**eparation (FGCN-DKS) in detail, which collaboratively solves the issue of consensus failure both within and across clients. Its core idea is to decouple knowledge that either promotes or hinders consensus, and then share the cluster-oriented, high-affinity components to regulate the guidance signals for each cluster accurately. As illustrated in Fig. 2, FGCN-DKS consists of three key modules: local pattern separation mechanism, common knowledge sharing strategy, and two-stage $K$-means clustering. The overall process is detailed in Algorithm 1.

### 3.1 NOTATIONS

We consider a non-IID federated setting with $N_c$ clients, where each client $i \in \{1, \ldots, N_c\}$ holds a private graph dataset containing $N_\phi$ clusters and $N_G$ graphs, denoted as $\mathcal{G} = \{G_j\}_{j=1}^{N_G}$. All datasets in the federated setting contain $N_\psi$ clusters. For each client, the node feature matrix is represented as $\mathbf{X} \in \mathbb{R}^{N \times d}$, and the normalized adjacency matrix is represented as $\mathbf{A} \in \{0,1\}^{N \times N}$, where $N$ is

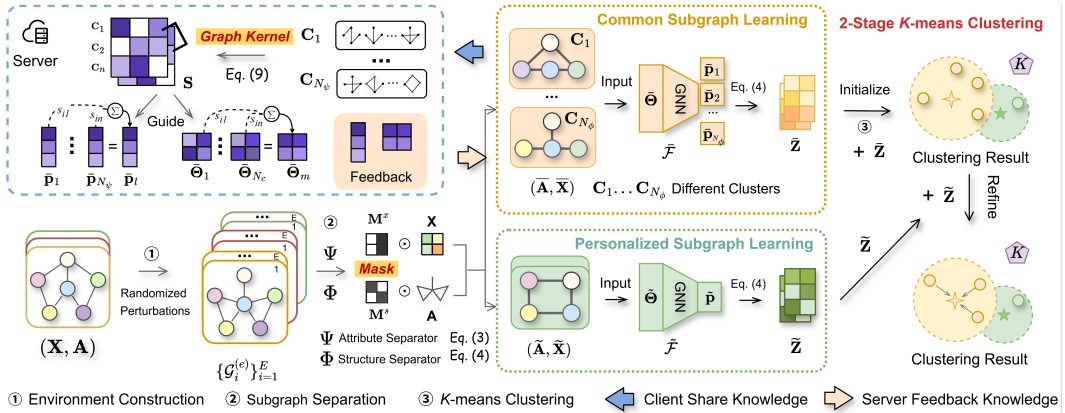

Figure 2: The framework of FGCN-DKS. We decouple the graph into invariant subgraphs and variable subgraphs, guided by clusters and clients, respectively. The invariant component is optimized in cluster-oriented coordination with global sharing, while the variable component further refines the clustering objectives. The two promote each other to produce clearer cluster boundaries.

the number of nodes, and $d$ is the dimension of node attributes. The total number of edges is denoted as $|\mathcal{E}|$. A detailed list of symbols is provided in the **Appendix** A.

### 3.2 LOCAL PATTERN SEPARATION MECHANISM

In this section, we attempt to decouple two distinctive subgraph patterns (i.e., common subgraph and personalized subgraph). Then, the knowledge stemming from the former is uploaded to the server for sharing, while the knowledge stemming from the latter is retained locally. To facilitate this, we define a series of environments by cluster, where each cluster represents a distinct distribution of graph data. The goal is to learn the invariant signal that characterizes the common pattern of each cluster while distinguishing the variant signal that reflects the personalized pattern of graphs. Specifically, for each original graph $\mathcal{G}_i$, we generate $E = N_\phi$ perturbed views of its graphs $\{\mathcal{G}_i^{(e)}\}_{e=1}^E$ through randomized structure and attribute perturbations to simulate distributional shift for each client. Each perturbed set $\mathcal{E}_k = \{\mathcal{G}_j^{(k)}\}_{j=1}^{N_G}$ defines an environment with graph structure $\mathbf{A}^{(k)}$.

**Subgraph Separation** To disentangle invariant and variant parts from each graph, we introduce node attribute separator $\Phi$, and graph structure separator $\Psi$, which generate the structure mask $\mathbf{M}^s$ and node attribute mask $\mathbf{M}^x$ for each graph $\mathcal{G}_i$, as:

$$\mathbf{M}^s = \Phi(\{\mathbf{A}^{(e)}\}_{e=1}^E, \mathbf{X}), \quad \mathbf{M}^x = \Psi(\{\mathbf{A}^{(e)}\}_{e=1}^E, \mathbf{X}). \tag{1}$$

Applying these masks yields two complementary adjacency matrices and node feature matrices, as

$$\bar{\mathbf{A}} = \mathbf{M}^s \odot \mathbf{A}, \quad \widetilde{\mathbf{A}} = (\mathbf{1} - \mathbf{M}^s) \odot \mathbf{A}, \tag{2}$$

$$\bar{\mathbf{X}} = \mathbf{M}^x \odot \mathbf{X}, \quad \widetilde{\mathbf{X}} = (\mathbf{1} - \mathbf{M}^x) \odot \mathbf{X}, \tag{3}$$

where $\odot$ denotes the Hadamard product. Thus, two subgraphs for any graph are obtained: $\bar{\mathcal{G}} = \{\bar{\mathbf{X}}, \bar{\mathbf{A}}\}$ and $\widetilde{\mathcal{G}} = \{\widetilde{\mathbf{X}}, \widetilde{\mathbf{A}}\}$. We then employ dual projector $\bar{\mathcal{F}}_\theta$ and $\widetilde{\mathcal{F}}_\theta$ based on GNN to extract node invariant features $\bar{\mathbf{H}}$ and variant node features $\widetilde{\mathbf{H}}$, as

$$\bar{\mathbf{H}} = \bar{\mathcal{F}}_\theta(\bar{\mathbf{A}}, \bar{\mathbf{X}} \mid \bar{\Theta}), \quad \widetilde{\mathbf{H}} = \widetilde{\mathcal{F}}_\theta(\widetilde{\mathbf{A}}, \widetilde{\mathbf{X}} \mid \tilde{\Theta}), \tag{4}$$

where $\bar{\Theta}$ and $\tilde{\Theta}$ are parameters of $\bar{\mathcal{F}}_\theta$ and $\widetilde{\mathcal{F}}_\theta$, respectively. Finally, graph-level representations $\bar{\mathbf{Z}}$ and $\tilde{\mathbf{Z}}$ are obtained via a READOUT function, as $\mathbf{Z} = \text{READ}(\mathbf{H})$. To encourage the projector to derive decoupled representations that are both cluster-discriminative and environment-invariant, we first enforce samples belonging to the same cluster to share similar invariant subgraphs. Given the set $\mathcal{P}_k = \{i \mid c(i) = k\}$ for cluster $k$, we minimize the pairwise variance within each group:

$$\mathcal{L}_{\text{inv}} = \sum_{ei=1}^E \sum_{ej=1}^E \sum_{k=1}^{N_\phi} \frac{1}{|\mathcal{P}_k|^2} \sum_{i,j \in \mathcal{P}_k} \|\bar{\mathbf{z}}_i^{(ei)} - \bar{\mathbf{z}}_j^{(ej)}\|^2. \tag{5}$$

To prevent the collision of invariant subgraphs from different clusters, which would hinder the capture of sufficiently distinguishable cluster features, we design $\mathcal{L}_{\text{div}}$ to increase the distance between invariant subgraphs from different clusters. Let $\mathcal{N} = (i, j) \mid c(i) \neq c(j)$ denote inter-cluster pairs, the $\mathcal{L}_{\text{div}}$ can be calculated as:

$$\mathcal{L}_{\text{div}} = \frac{1}{|\mathcal{N}|} \sum_{ei=1}^{E} \sum_{ej=1}^{E} \sum_{k=1}^{N_c} \sum_{(i,j) \in \mathcal{N}} \vartheta(\mathbf{z}_i^{(ei)}, \bar{\mathbf{z}}_j^{(ei)}), \tag{6}$$

where $\vartheta(\cdot, \cdot)$ is a inverse distance function. This term explicitly encourages inter-cluster separation in the embedding space. To enforce invariance under environmental shifts, we minimize the variation of each graph's invariant representation across different environments, as:

$$\mathcal{L}_{\text{env}} = \frac{1}{EN_\phi} \sum_{i=1}^{N_\phi} \sum_{e=1}^{E} \|\mathbf{z}_i^{(e)} - \bar{\mathbf{z}}_i\|^2 + \frac{1}{E} \sum_{e=1}^{E} \vartheta(\bar{\mathbf{Z}}^{(e)}, \tilde{\mathbf{Z}}^{(e)}), \tag{7}$$

where $\bar{\mathbf{z}}_i = \frac{1}{E} \sum_{e=1}^{E} \mathbf{z}_i^{(e)}$. This objective encourages the encoder to focus on information that is stable across distributional shifts, enhancing generalization to unseen environments, while effectively separating the invariant and variant components, ensuring the model retains the stable structure of the graph while capturing the patterns of variation. Finally, the overall optimization objective $\mathcal{L}$ in each client is as:

$$\mathcal{L} = \mathcal{L}_{\text{inv}} + \beta \mathcal{L}_{\text{div}} + \gamma \mathcal{L}_{\text{env}} + \mathcal{L}_{\text{mse}}, \tag{8}$$

where $\mathcal{L}_{\text{mse}}$ is the node representation reconstruction loss. $\beta$ and $\gamma$ are two hyperparameters that control the ratio of the loss. By doing so, we separate each graph into two distinct subgraph patterns (i.e., variant subgraph and variant subgraph). The theoretical and experimental effectiveness of the subgraph separation process are presented in **Appendix** B and **Appendix** C, respectively. Subsequently, these common subgraph structures $\bar{\mathbb{G}}$ represented as $\mathbf{C}$ are uploaded and serve as irrecoverable digest information on the server to reflect inter-cluster affinity, facilitating the achievement of personalized consensus. Meanwhile, the invariant subgraphs are kept on the client, offering crucial guidance for clustering while safeguarding privacy. This design strikes a balance by enabling the global model parameters to maintain coherence in local knowledge while simultaneously adapting to local distributional variables.

## 3.3 COMMON KNOWLEDGE AGGREGATION STRATEGY

In this section, we design a Common Knowledge Sharing Strategy (CKSS), aiming to aggregate negotiated-friendly knowledge at a finer level of granularity, mitigating the impact of weak correlations on the target and enhancing the overall quantity of global consensus. First, the server receives the common prototype, parameters, and pattern digests from the clients. Next, we employ cluster-oriented common pattern digests derived from subgraphs to capture the relation of stable structural semantics across clients. Finally, we leverage these relations to guide the aggregation of parameters and prototypes, achieving personalized knowledge consensus for different clients.

**Cluster-oriented Information Aggregation** Since the pattern digests reflect the underlying manifold structure of the cluster, we utilize the cluster-oriented pattern digest uploaded from each client to compute potential relationships using graph kernels, such as RW (Kang et al., 2012), WL (Liu et al., 2025b), SP (Borgwardt et al., 2020), LT (Johansson et al., 2014), and others. These graph kernels effectively capture the similarity between cluster patterns, aligning local information within each cluster with the global structure, while ensuring privacy protection. Specifically, the similarity between cluster $i$ and $j$ is given by $k(\mathbf{C}_i, \mathbf{C}_j)$. The pairwise affinity matrix $\mathbf{S}$ is then computed as: $\mathbf{S}_{ij} = k(\mathbf{C}_i, \mathbf{C}_j)$, where $k(\cdot, \cdot)$ is the graph kernel method and $\mathbf{C}_i$ is the pattern digest from $i$-th cluster. In this way, we obtain the initialized affinity matrix of all clusters. However, relying solely on this initialization relationship to propagate knowledge is insufficient; we also need to incorporate additional information to ensure a more robust aggregation process. Therefore, we further introduce historical information to define a stability coefficient $\alpha$ to quantify the stability of relationships between clusters over multiple iterations, as

$$\alpha_{ij} = \frac{|k(\mathbf{C}_i^{(t)}, \mathbf{C}_j^{(t)}) - k(\mathbf{C}_i^{(t-1)}, \mathbf{C}_j^{(t-1)})|}{\max(k(\mathbf{C}_i^{(t)}, \mathbf{C}_j^{(t)}), \epsilon)}, \tag{9}$$

where $t$ is the communication epoch. A smaller $\alpha_{ij}$ indicates a more stable relationship between clusters. Then the affinity matrix can be updated as

$$\mathbf{S}^{(t)} = (1 - \lambda) \cdot \mathbf{S}^{(t-1)} + \lambda \cdot \sum_{i,j} \alpha_{ij} \cdot k(\mathbf{C}_i^{(t)}, \mathbf{C}_j^{(t)}), \tag{10}$$

where $\lambda$ is a hyperparameter controlling the relative weight between historical and current similarity information. This smoothing trick ensures that the similarity matrix evolves gradually over iterations, avoiding over-reliance on single-round updates, and stabilizing the convergence process. Subsequently, the server performs personalized aggregation based on the cluster affinity to obtain a consensus guide signal. The consensus prototype $\bar{\mathbf{p}}_{glo|l}$ for cluster $l$ and the consensus parameters $\bar{\Theta}_{glo|m}$ for client $m$ are calculated as:

$$\bar{\mathbf{P}}_{glo|l} = \sum_{i=1}^{N_\psi} s_{li} \cdot \tilde{\mathbf{p}}_i, \quad \bar{\Theta}_{glo|m} = \sum_{j \in \mathcal{S}_m} \sum_{u=1}^{N_\psi} s_{uj} \bar{\Theta}_u, \tag{11}$$

where $\mathcal{S}_m$ denotes cluster set from client $m$. It is noteworthy that this alignment scheme differs from traditional methods, which require equal cluster quantities for proportional division. Instead, it leverages the inherent affinity of clusters through pattern relationships, guiding clients to delineate clearer clustering boundaries.

This strategy allows each client to benefit from similar peers in the same latent space, while avoiding negative transfer from unrelated distributions. Compared to the naive average strategy, our method explores the relationship between patterns, allowing clients to be guided by more personalized knowledge with greater affinity, enabling them to exert greater clustering advantages.

### 3.4 Two Stage K-means Clustering

When the personalized consensus knowledge is generated and fed back to the local models for optimization, we further exploit the disentangled representations learned through invariant training by introducing a two-stage clustering. This process first captures cluster-oriented stable patterns and then refines client-oriented personalized information. Specifically, we first perform clustering over the invariant representations $\bar{\mathbf{Z}}$ using a standard $K$-means algorithm. Since these representations are learned to be robust against environment-specific perturbations, the initial clustering $\mathcal{C}^{(0)}$ provides a reliable global semantic grouping. Then, we further refine the initial clusters by leveraging the variant representations $\tilde{\mathbf{Z}}$, which are specifically designed to encode environment-sensitive or instance-level information. Within each initial cluster $\mathcal{C}_k^{(0)}$, we perform a secondary clustering or similarity-based refinement to

---

**Algorithm 1** Algorithm Procedure of FGCN-DKS

**Require:** Initial model parameters $\{\bar{\Theta}_i\}_{i=1}^{N_c}$, Node feature $\mathbf{X}$, Adjacent matrix $\mathbf{A}$, Client number $N_c$.
**Ensure:** Clustering Result $R$.
1: on each client
2: **for** $c = 1 \rightarrow N_c$ **do**
3:     Generate $E$ perturbed graphs $\{\mathcal{G}^{(e)}\}_{e=1}^E$ to construct environments.
4:     Obtain invariant mask $\mathbf{M}^s$ and $\mathbf{M}^x$ by Eq. (2).
5:     Separate two type subgraphs $\bar{\mathbf{A}}$ and $\tilde{\mathbf{A}}$ by Eq. (3).
6:     Extract dual embeddings $\tilde{\mathbf{Z}}$ and $\bar{\mathbf{Z}}$ by Eq. (4).
7:     Upload common prototype $\bar{\mathbf{p}}$, pattern digest $\mathbf{C}$ and invariant encoder parameters $\bar{\Theta}$ to the server.
8: **end for**
9: on the Server
10: Collect $\mathbf{C}$, $\bar{\mathbf{p}}$ and $\bar{\Theta}$ from each client to the server.
11: Calculate affinity matrix $\mathbf{S}$ by Eqs. (9) - (10).
12: Personalized aggregate $\{\bar{\mathbf{p}}_i\}_{i=1}^{N_\psi}$ and $\{\bar{\Theta}_i\}_{i=1}^{N_c}$ to generate consensus knowledge by Eq. (11).
13: Feedback parameters and prototype to each client.
14: Execute 2-stage $K$-means clustering.
15: **return** $R$

---

enhance the granularity and expressiveness of the final partitioning. This variant-aware refinement step enables the model to adaptively adjust for intra-cluster diversity, thereby improving clustering fidelity and interpretability. Overall, this common-to-personalized clustering paradigm enables a robust yet flexible representation-driven grouping mechanism. The invariant component ensures cross-environment consistency, while the variant component captures local distinctions, jointly facilitating high-quality cluster assignments even under distributional shifts.

### 3.5 Efficiency Analysis

Compared with the standard parameter averaging in FedAvg, our framework introduces only a slight increase in global computation through affinity-guided consensus aggregation. FedAvg performs a weighted average with complexity $\mathcal{O}(d^2)$, whereas our method additionally computes cluster-level affinities from pattern digests at $\mathcal{O}(N_\psi^2 \kappa)$, where $N_\psi \ll d$ and $\kappa$ is any linear kernel in practice. The

subsequent personalized aggregation requires only $\mathcal{O}(N_c N_\psi d)$ complexity. Therefore, the increase in computational complexity is acceptable given the corresponding performance gains.

| Models | SM²(7) | | | | SN³(2) | | | | SM-BIO²(9) | | | |
|---|---|---|---|---|---|---|---|---|---|---|---|---|
| | ACC | NMI | ARI | F1 | ACC | NMI | ARI | F1 | ACC | NMI | ARI | F1 |
| FedSage* | 55.6±1.4 | 12.2±1.3 | 7.6±0.6 | 50.2±1.0 | 53.3±1.9 | 14.8±1.4 | 11.6±2.8 | 49.3±2.0 | 57.4±2.2 | 5.2±2.1 | 4.2±2.7 | 49.9±0.5 |
| GCFL* | 61.1±1.7 | 8.7±2.4 | 9.4±2.4 | 49.6±2.3 | 52.1±2.3 | 12.5±2.3 | 13.2±2.3 | 52.3±1.6 | 60.1±1.8 | 4.7±2.4 | 3.2±2.3 | 47.3±1.5 |
| FedStar* | 58.9±2.4 | 12.0±1.2 | 0.1±0.8 | 49.7±2.8 | 51.7±2.7 | 13.7±2.8 | 12.4±1.9 | 50.7±2.3 | 59.5±1.6 | 5.3±1.5 | 3.8±2.0 | 51.7±2.2 |
| LG-FGAD† | 65.8±0.8 | 18.8±1.9 | 3.4±1.1 | 62.9±0.6 | 37.9±2.5 | 9.6±2.9 | 0.4±0.7 | 26.3±3.5 | 59.6±1.5 | 9.0±1.4 | 7.8±1.7 | 56.0±2.6 |
| FGAD† | 66.4±2.3 | 20.2±2.6 | 4.3±3.2 | 63.8±2.6 | 41.2±1.9 | 5.8±2.4 | 0.5±1.4 | 35.8±1.6 | 63.5±1.0 | 14.7±1.1 | 2.1±2.0 | 60.7±1.5 |
| AGDiff† | 70.2±1.4 | 19.3±2.9 | 15.6±3.9 | 67.3±2.8 | 42.3±1.9 | 7.5±1.2 | 8.6±2.0 | 37.2±1.0 | 61.3±2.5 | 10.6±1.9 | 3.4±0.2 | 57.2±1.6 |
| GLCC‡ | 56.2±2.8 | 8.6±3.4 | 5.4±4.2 | 53.7±4.1 | 43.5±2.0 | 9.7±2.4 | 3.5±1.5 | 40.7±2.1 | 57.5±2.3 | 6.7±1.8 | 4.6±2.0 | 41.6±1.5 |
| UDGC‡ | 53.6±3.4 | 9.7±2.5 | 8.6±3.4 | 53.4±2.3 | 50.1±2.4 | 10.4±2.5 | 9.3±1.4 | 48.4±2.3 | 55.6±1.8 | 8.9±1.4 | 6.8±0.4 | 50.4±1.0 |
| DGLC‡ | 60.8±1.5 | 14.3±1.2 | 10.7±1.4 | 52.2±1.4 | 55.5±1.5 | 11.6±2.8 | 12.3±1.7 | 52.1±2.1 | 58.0±1.9 | 12.3±1.4 | 11.6±1.7 | 53.5±1.5 |
| DCGLC‡ | 63.1±1.7 | 17.5±1.5 | 17.6±1.7 | 58.4±2.0 | 59.6±1.9 | 13.7±2.0 | 15.6±1.8 | 56.8±2.3 | 60.4±1.6 | 13.2±1.1 | 15.8±1.9 | 56.6±1.2 |
| FedGCN | 75.9±0.8 | 23.1±1.6 | 31.3±3.4 | 67.1±1.5 | 66.6±2.3 | 30.4±6.6 | 34.1±5.3 | 50.8±2.4 | 69.2±0.6 | 14.0±2.7 | 17.5±3.1 | 59.1±0.9 |
| FedPKA | 77.0±0.2 | 26.8±3.8 | 31.2±3.3 | 67.3±2.0 | 67.5±1.5 | 25.7±2.3 | 32.6±2.4 | 55.5±1.5 | 70.8±1.4 | 15.4±2.6 | 19.6±3.4 | 60.6±2.1 |
| **OURS** | **79.2±0.5** | **28.3±1.1** | **34.6±0.9** | **72.3±1.1** | **70.2±0.4** | **34.2±1.7** | **36.8±1.2** | **60.4±1.9** | **73.8±2.2** | **17.7±2.4** | **21.3±2.0** | **61.3±1.7** |
| | SM-BIO-SY²(10) | | | | SN-SY¹¹(2) | | | | CV¹⁵(3) | | | |
| FedSage* | 57.6±1.9 | 20.6±1.9 | 17.6±2.4 | 46.7±1.8 | 15.6±1.1 | 7.6±1.0 | 3.4±2.7 | 2.9±1.8 | 19.6±0.8 | 22.7±0.4 | 12.5±1.3 | 18.2±0.8 |
| GCFL* | 59.1±2.0 | 14.4±2.2 | 13.7±2.8 | 52.3±1.9 | 19.3±0.6 | 4.5±2.3 | 1.2±1.1 | 8.7±0.9 | 27.9±1.6 | 27.5±2.2 | 13.1±1.9 | 27.4±1.3 |
| FedStar* | 57.9±2.6 | 15.7±2.4 | 16.1±3.0 | 52.3±2.2 | 19.0±2.9 | 4.1±2.5 | 2.3±2.5 | 7.9±2.2 | 22.7±1.0 | 20.3±1.7 | 10.3±2.1 | 20.2±3.2 |
| LG-FGAD† | 58.4±0.5 | 7.6±0.4 | 6.4±0.8 | 54.6±0.7 | 19.1±1.4 | 6.7±0.6 | 3.3±1.0 | 7.4±1.1 | 27.4±1.6 | 31.4±4.0 | 9.6±3.2 | 24.7±3.3 |
| FGAD† | 62.2±1.2 | 14.6±2.6 | 3.0±2.9 | 56.7±0.8 | 16.4±0.5 | 7.4±0.5 | 2.7±0.3 | 8.3±0.7 | 26.0±1.1 | 31.9±1.2 | 7.3±1.1 | 25.2±1.1 |
| AGDiff† | 61.4±1.3 | 15.6±2.8 | 13.5±1.4 | 50.4±3.0 | 15.8±2.0 | 4.3±2.0 | 3.4±0.2 | 9.6±2.8 | 23.6±1.4 | 27.5±1.3 | 8.7±0.9 | 22.8±1.4 |
| GLCC‡ | 54.2±3.5 | 10.8±1.3 | 7.6±0.9 | 53.5±1.6 | 16.3±2.6 | 3.8±2.1 | 3.2±2.0 | 10.0±2.3 | 22.8±1.2 | 20.4±1.3 | 10.6±1.5 | 14.2±1.0 |
| UDGC‡ | 55.6±2.9 | 12.7±2.4 | 11.4±2.6 | 54.1±2.2 | 17.5±1.2 | 8.1±2.2 | 6.4±3.5 | 9.7±2.9 | 20.4±2.3 | 10.5±2.3 | 8.2±1.6 | 14.7±1.8 |
| DGLC‡ | 57.8±2.0 | 14.4±1.3 | 10.7±1.6 | 54.3±1.2 | 18.2±1.0 | 9.1±1.4 | 7.5±1.1 | 8.6±1.3 | 29.5±2.4 | 21.6±1.3 | 14.5±1.4 | 22.1±1.7 |
| DCGLC‡ | 60.1±1.4 | 15.6±1.7 | 13.1±1.2 | 59.7±1.8 | 17.5±1.2 | 8.2±1.4 | 6.5±2.0 | 10.4±2.6 | 28.8±2.0 | 24.3±1.1 | 18.6±1.2 | 24.5±1.3 |
| FedGCN | 68.6±1.3 | 13.5±2.1 | 17.2±3.6 | 59.4±3.8 | 18.3±3.1 | 4.8±5.0 | 2.3±2.6 | 11.2±3.5 | 34.6±2.8 | 34.8±2.4 | 19.3±2.3 | 31.6±2.9 |
| FedPKA | 70.1±0.9 | 17.2±0.8 | 22.2±1.1 | 61.5±2.3 | 16.4±2.6 | 5.7±2.3 | 5.9±2.0 | 8.2±2.5 | 36.4±1.1 | 34.4±1.6 | 20.3±1.2 | 33.5±1.3 |
| **OURS** | **73.6±1.4** | **22.7±1.2** | **23.5±1.9** | **64.4±1.7** | **23.5±1.5** | **13.4±1.0** | **8.7±1.6** | **15.6±1.2** | **39.2±1.3** | **37.1±1.5** | **24.5±1.2** | **35.2±1.4** |

Table 2: Performance comparison across different FGL methods under six non-IID settings. * denotes supervised methods adapted for unsupervised learning. † denotes anomaly detection methods adapted for clustering. ‡ denotes centralized deep graph-level clustering methods adapted for FGL.

# 4 EXPERIMENTS

In this section, we conduct extensive experiments to evaluate the effectiveness and robustness of FGCN-DKS. We first introduce the experimental setup, including datasets, baseline methods, and implementation details. Then, we present the comparison results with the SOTA approaches, followed by ablation studies to examine the contribution of each component. Finally, we provide additional analysis further to validate the efficiency and generalizability of our methods.

## 4.1 EXPERIMENT SETTINGS

**Benchmark Datasets** We evaluate FGCN-DKS on 16 public datasets from the TUDataset collection, spanning small molecules, bioinformatics, computer vision, and social networks (Liang et al., 2025). These datasets are organized into six non-IID settings: same-domain (SM, SN, CV) and cross-domain (SM-BIO, SM-BIO-SY, SN-SY), aligned following FedGCN (Liu et al., 2025a). Detailed dataset information and non-IID settings are provided in **Appendix** D.

**Evaluation Metrics** To comprehensively assess the performance of FGCN-DKS, we adopt four standard unsupervised clustering evaluation metrics: Accuracy (ACC) (Cai et al., 2022; 2024a), Normalized Mutual Information (NMI) (Liang et al., 2024b), Adjusted Rand Index (ARI) (Cai et al., 2024b), and F1 Score (Tu et al., 2024). Details of these metrics are given in the **Appendix** E.

**Baseline Methods** To evaluate the effectiveness of FGCN-DKS, we consider two types of baselines. The first includes the SOTA FGC method FedGCN, FedPKA, and several representative FGL methods adapted to FGC, such as FedSage (Zhang et al., 2021), GCFL (Xie et al., 2021), FedStar (Tan et al., 2023), LG-FGAD (Cai et al., 2024c), FGAD (Cai et al., 2024d), and AGDiff (Cai et al., 2025). The second comprises advanced centralized Deep Graph-level Clustering (DGC) methods,

including GLCC (Ju et al., 2023), UDGC (Hu et al., 2023), DGLC (Cai et al., 2024a), and DCGLC (Cai et al., 2024b). As these models rely on full data access, we adapt them to the FGL scenario by local training and parameter aggregation, ensuring fair comparison. The description of the baselines and the implementation details of FGCN-DKS are provided in **Appendix** F.

## 4.2 Comparison Experiments

**Comparison with FGL Methods** We compare FGCN-DKS with advanced FGL methods to assess the performance. As illustrated in Table 2, the experimental results lead to the following observations: Our approach delivers superior performance, primarily because invariant learning effectively disentangles the two structural patterns and aggregates them globally in a cluster-oriented manner. This process refines cluster-relevant signals while preserving the consistency of invariant representation learning. Compared with supervised FGC methods and unsupervised anomaly detection approaches, our method achieves superior performance. Supervised methods rely on label guidance, and without labels, they lack a reliable signal to define meaningful cluster boundaries, leading to degraded clustering quality, while unsupervised anomaly detection focuses on identifying rare, distinctive graph patterns rather than general clustering. Moreover, compared with SOTA FGC methods, our approach still demonstrates a significant advantage, indicating that merely sharing abundant parameters and prototypes does not necessarily lead to more effective performance improvement.

**Comparison with Centralized DGC Methods** To further assess the performance of FGCN-DKS, we compare it with some representative centralized DGC methods. The experimental results are shown in Table 2, and the following conclusions are obtained: Compared with existing advanced methods, FGCN-DKS significantly improves performance, which is mainly attributed to the lack of ability of existing methods to perceive cluster-directed signal to adjust learning strategies. The inherent paradigm will cause consensus failure due to large differences in optimization directions between parameters. In contrast, our method cleverly separates the two structural patterns and uses the cluster summary as a prototype to guide the server to learn with different strategies, overcoming the difference in semantic granularity and improving the overall performance of the model.

**Comparison with Supervised FGL Methods** To further demonstrate the superior performance of FGCN-DKS, we conduct a comparison with several supervised methods by providing them with a partial set of labels. The experimental results are shown in Fig. 3, which leads to the following conclusions: Compared with supervised methods, our method still shows strong performance despite the lack of labels. This is mainly attributed to the fact that our method can effectively separate cluster-driven knowledge and personalized features locally and use different strategies to aggregate cluster-friendly guidance signals, improving the global performance.

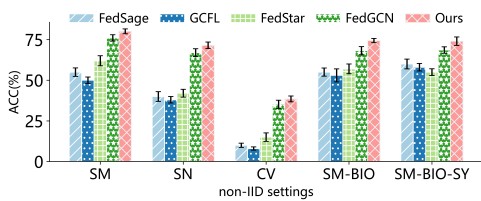

Figure 3: Comparison experiment results on the supervised methods with few labels under five non-IID settings.

## 4.3 Module Ablation Studies

To provide a clearer and more systematic understanding of how each component contributes to the overall performance of our framework, we organize the ablation settings into four representative variants. The first adopts only the minimal client and server settings, forming a basic baseline without any advanced mechanisms (**Basic**). The second removes the subgraph pattern separation module and the two-stage $k$-means refinement, while retaining the basic local learning strategy and keeping the server unchanged (**-Local**). The third replaces CKSS with the standard FedAvg aggregation while preserving the complete local inference and learning pipeline (**-Server**). The final activates the full proposed framework, where all modules and optimization mechanisms are jointly enabled (**Ours**). This structured decomposition allows a fine-grained quantification of the importance of each design choice. The experiment results are shown in Table 3, which reveal several notable observations. First, performing knowledge separation solely on the client side already yields consistent improvements across all datasets. This indicates that mitigating local knowledge heterogeneity plays a crucial role in obtaining clearer representations of both shared and personalized graph pat-

| Variants | SM | | | SM-BIO | | | SM-BIO-SY | | | CV | | |
|---|---|---|---|---|---|---|---|---|---|---|---|---|
| | ACC | NMI | ARI | ACC | NMI | ARI | ACC | NMI | ARI | ACC | NMI | ARI |
| Basic | 61.7±1.2 | 19.5±1.6 | 14.5±1.4 | 59.3±2.1 | 15.2±1.6 | 13.8±2.1 | 56.3±2.9 | 7.7±2.4 | 13.3±1.9 | 29.5±2.4 | 18.6±1.8 | 16.3±1.4 |
| -Local | 64.6±1.4 | 22.4±1.3 | 16.9±1.7 | 61.6±2.0 | 17.9±2.1 | 16.2±1.7 | 58.4±2.9 | 8.9±2.5 | 15.7±1.8 | 32.4±2.3 | 20.4±1.9 | 19.7±2.0 |
| -Server | 68.1±1.9 | 23.9±2.1 | 32.0±1.6 | 69.5±1.5 | 17.4±2.3 | 19.2±1.6 | 67.2±1.3 | 16.5±1.2 | 19.6±1.5 | 37.7±1.0 | 33.5±0.7 | 22.6±0.7 |
| Ours | 79.2±0.5 | 28.3±1.1 | 34.6±0.9 | 73.8±2.2 | 17.7±2.4 | 21.3±2.0 | 73.6±1.4 | 22.0±1.2 | 23.5±1.9 | 39.2±1.3 | 37.1±1.6 | 24.5±1.3 |

Table 3: Module ablation study results on SM, SM-BIO, SM-BIO-SY, and SN non-IID settings.

| Clients | Ours | | FedGCN | | FedPKA | | FedAvg | |
|---|---|---|---|---|---|---|---|---|
| | Time (s) | Cost (KB) | Time (s) | Cost (KB) | Time (s) | Cost (KB) | Time (s) | Cost (KB) |
| 1 | 16.8 | 32.3 | 15.9 | 30.5 | 20.6 | 42.1 | 14.6 | 28.7 |
| 2 | 35.5 | 67.3 | 30.8 | 61.8 | 41.5 | 85.4 | 29.1 | 57.5 |
| 3 | 53.0 | 96.7 | 44.8 | 93.6 | 64.4 | 131.6 | 45.6 | 86.8 |

Table 4: Communication overhead comparison under the CV non-IID setting.

terns. Second, using CKSS alone brings only moderate gains. This limitation arises because CKSS fundamentally relies on reliable common subgraph patterns extracted through local separation; without them, the server struggles to accurately estimate inter-client affinities, reducing the effectiveness of global consensus modeling. Finally, when both modules operate jointly, they reinforce each other, leading to substantial improvements in clustering performance. These results collectively demonstrate that local knowledge disentanglement and global consensus optimization are complementary and jointly necessary for achieving high-quality federated graph-level clustering.

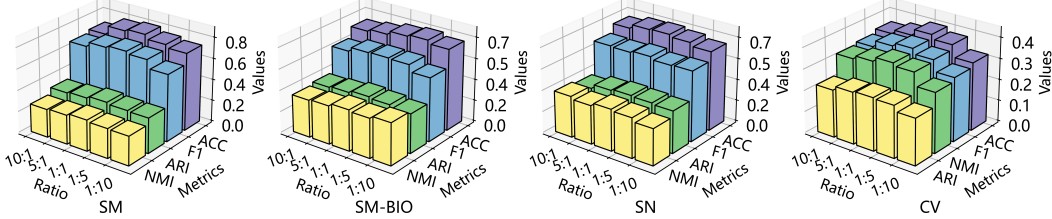

Figure 4: Hyper-parameters $\alpha$ and $\beta$ sensitivity analysis results under four non-IID settings with varying $\alpha$:$\beta$ ratios in the range of [1:10, 10:1], reporting ACC, NMI, ARI and F1 values.

## 4.4 HYPER-PARAMETERS SENSITIVITY ANALYSIS

To investigate the effects of each loss component in Eq. (8), we perform a sensitivity analysis by varying the weighting hyperparameters $\alpha$ and $\beta$. The experimental results are shown in Fig. 4, and the following conclusions are obtained: FGCN-DKS achieved optimal balanced performance at a 1:1 ratio. Increasing the ratio slightly improved NMI and ARI, but led to a decline in ACC and F1. Conversely, decreasing the ratio exhibited the opposite trend.

## 4.5 COMMUNICATION OVERHEAD ANALYSIS

To further evaluate the practicality of the proposed framework in federated environments, we conduct a communication overhead analysis. As shown in Table 4, our method incurs slightly higher communication time and communication payload than FedGCN. This increase mainly results from transmitting shared structural patterns, and the overall additional cost remains negligible relative to the full model parameters. Compared with FedPKA, our framework shows clear advantages in both communication time and communication payload. FedPKA requires frequent model gradient exchanges, which substantially increases its communication burden. In contrast, our design effectively reduces unnecessary transmissions while preserving model performance. We also compare our approach with the standard FedAvg baseline. Although additional structure patterns are transmitted, their size is small. The resulting communication overhead remains marginal and does not affect

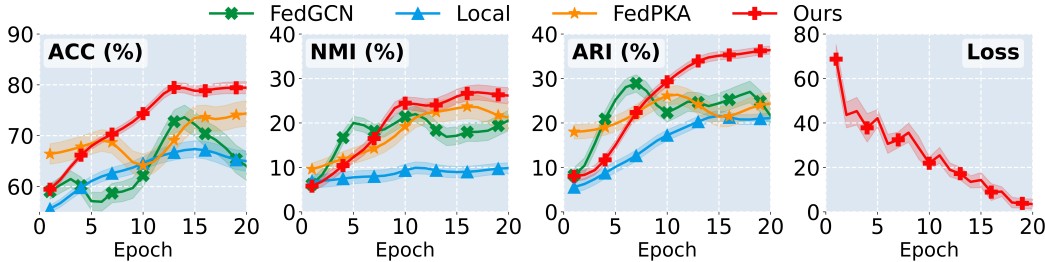

Figure 5: Convergence curves on ACC, NMI, ARI clustering metrics and loss values under SM non-IID setting, compared with FedPKA, FedGCN federated graph-level clustering methods.

the overall efficiency of the model. Meanwhile, this lightweight increase brings notable performance gains. Overall, the communication analysis indicates that our method achieves competitive efficiency while maintaining strong performance across heterogeneous federated settings.

## 4.6 CONVERGENCE STUDIES

To assess the stability and convergence behavior of FGCN-DKS, we track the trajectories of ACC, NMI, ARI, and loss values across communication rounds and compare them with FedGCN and FedPKA. As shown in Fig. 5, our model converges smoothly and rapidly with stable performance on all metrics. Although FedPKA shows relatively strong early-stage performance due to its community division mechanism, it fails to maintain improvement and ultimately does not converge. In addition, only local training also converges, but remains clearly inferior without server coordination. The training loss decreases steadily with only minor fluctuations, indicating robust optimization dynamics under federated settings. Overall, these results confirm that FGCN-DKS achieves reliable and stable convergence throughout the training process.

## 4.7 CLIENT-WISE PERFORMANCE COMPARISON

To further evaluate the client-level effectiveness of FGCN-DKS, we conduct a client-wise performance comparison, as shown in Fig. 6. The results show that our FGL method consistently improves the accuracy of all clients, demonstrating strong robustness under heterogeneous data distributions. Although FedGCN enhances the overall performance to some extent, it does so while reducing the accuracy of several clients, indicating an unbalanced aggregation effect. In contrast, our approach yields both global performance gains and stable client-level improvements, thereby achieving a more reliable and uniformly beneficial optimization across all participants.

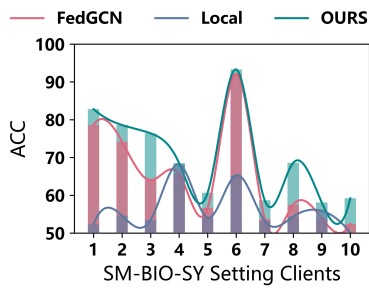

Figure 6: Client-wise performance comparison experiment results under SM-BIO-SY non-IID setting.

## 5 CONCLUSION

In this paper, we propose FGCN-DKS, a federated clustering framework that effectively addresses the challenge of consensus failure caused by knowledge heterogeneity. By improving invariant learning and common knowledge shared strategy, our method decouples on two levels: (1) shared subgraph patterns and personalized subgraph patterns, and (2) Cluster-oriented consensus pattern and client-driven prior knowledge negotiation. Through this elegant design, we upload only the shared subgraph pattern digests to the server for consensus optimization, focusing on the most beneficial parts for clustering, while the personalized subgraph patterns are retained locally to refine the clustering process by the 2-stage $K$-means clustering process. Regardless of the distribution pattern on the clients, our approach achieves superior performance compared to existing state-of-the-art methods. In the future, we plan to address this challenge at the node level, enabling more flexible clustering without being overly constrained by inherent priors.

## 6 ACKNOWLEDGMENTS

This work was supported by the National Natural Science Foundation of China (No. 62302333, 92370111, and 62272340) and the Open Research Fund from Guangdong Laboratory of Artificial Intelligence and Digital Economy (SZ) (No. GML-KF-24-16)

## 7 REPRODUCIBILITY STATEMENT

We provide all essential details, including datasets, pseudocodes, hyperparameters, and environment settings, to facilitate the reproducibility of our experiments.

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

## A    NOTATIONS

All notations used in the proposed FGCN-DKS are summarized in Table 5.

## B    GRAPH DECOMPOSITION INTO INVARIANT AND VARIANT SUBGRAPHS

PROBLEM SETUP AND NOTATION

Let $G = (V, E)$ be an undirected graph with node set $V$ and edge set $E$, where each node $v \in V$ has a feature vector $\mathbf{x}_v \in \mathbb{R}^d$. We assume that the graph $G$ can be decomposed into two components: an invariant subgraph $G_{\text{inv}}$ and a variant subgraph $G_{\text{var}}$, such that:

$$G = G_{\text{inv}} \cup G_{\text{var}}, \tag{12}$$

where $G_{\text{inv}}$ is the invariant subgraph that represents the part of the graph that remains unchanged across different datasets or transformations. $G_{\text{var}}$ is the variant subgraph that represents the part of the graph that is sensitive to changes in the data, such as variations across different experiments, graphs, or time steps.

Our goal is to provide a formal decomposition of the graph into these two subgraphs and to prove that this decomposition is meaningful in terms of graph properties.

### B.1    DECOMPOSITION METHODOLOGY

The decomposition process follows these steps:

1. **Identification of Invariant Components:** We first identify the subgraph $G_{\text{inv}}$ by looking for the parts of the graph that are consistent across all observed instances. This is done by comparing the graph's structure and node features across different datasets or views.

2. **Identification of Variant Components:** The remaining graph, $G_{\text{var}}$, consists of the components that change depending on external factors. These components are identified by measuring the variation in the graph's structure or node features over time or across different instances.

3. **Formalizing the Decomposition:** The graph $G$ is decomposed into two disjoint subgraphs $G_{\text{inv}}$ and $G_{\text{var}}$, such that:

$$G = G_{\text{inv}} \cup G_{\text{var}}, \quad G_{\text{inv}} \cap G_{\text{var}} = \emptyset. \tag{13}$$

The invariant subgraph captures the stable, core relationships in the graph, while the variant subgraph contains the dynamic or fluctuating parts.

| Notations | Meaning | Notations | Meaning |
|---|---|---|---|
| $N$ | Number of nodes | $N_G$ | Number of graphs from each client |
| $N_c$ | Number of clients | $N_\phi$ | Number of clusters from each client |
| $N_\psi$ | Number of clusters from all clients | $E$ | Number of environments |
| $d$ | Dimensions of node attribute | $d'$ | Dimensions of node embedding |
| $\mathbf{X}$ | Node attribute matrix | $\mathbf{A}$ | Graph adjacency matrix |
| $\mathbf{M}^s$ | Graph structure mask matrix | $\mathbf{M}^x$ | Node attribute mask matrix |
| $\bar{\mathbf{X}}$ | Invariant node attribute matrix | $\bar{\mathbf{A}}$ | Invariant graph adjacency matrix |
| $\tilde{\mathbf{X}}$ | Variant node attribute matrix | $\tilde{\mathbf{A}}$ | Variant graph adjacency matrix |
| $\bar{\mathbf{H}}$ | Invariant node embeddings | $\tilde{\mathbf{H}}$ | Variant node embeddings |
| $\bar{\mathbf{Z}}$ | Invariant graph-level embeddings | $\tilde{\mathbf{Z}}$ | Variant graph-level embeddings |
| $\bar{\mathbf{P}}$ | Common prototype | $\tilde{\mathbf{P}}$ | Personalized prototype |
| $\mathbf{P}_{glo}$ | Consensus prototype | $\mathbf{\Theta}_{glo}$ | Consensus parameter matrix |
| $\bar{\mathbf{\Theta}}$ | Invariant model parameter matrix | $\tilde{\mathbf{\Theta}}$ | Variant model parameter matrix |
| $\gamma$ | Weight hyper-parameter | $\beta$ | Weight hyper-parameter |
| $\mathbf{S}$ | Affinity matrix | $\alpha$ | Stability coefficient |
| $\mathcal{S}$ | Cluster set | $\mathcal{C}$ | Cluster node set |
| $\epsilon$ | A small lower bound constant | $\lambda$ | Weight hyper-parameter |

Table 5: Basic notations for the proposed FGCN-DKS.

INVARIANT AND VARIANT SUBGRAPH PROPERTIES

For the decomposition to be valid, we must ensure that the invariant subgraph captures only those parts of the graph that are consistent across multiple views or datasets. We define the following properties for the invariant and variant subgraphs:

- **Invariant Subgraph** ($G_{\mathbf{inv}}$): The invariant subgraph contains the edges and nodes that remain unchanged across different instances. Formally, for any two graphs $G_1 = (V_1, E_1)$ and $G_2 = (V_2, E_2)$ with the same node set $V$, the edges in $G_{\mathrm{inv}}$ must satisfy:

$$E_{\mathrm{inv}} \subseteq E_1 \cap E_2, \quad \forall E_1, E_2 \in \{E_1, E_2\}. \tag{14}$$

This ensures that the edges in $G_{\mathrm{inv}}$ are consistent across all graphs or datasets.

- **Variant Subgraph** ($G_{\mathbf{var}}$): The variant subgraph contains the edges and nodes that differ between graphs. This can be formally defined as:

$$E_{\mathrm{var}} = E \setminus E_{\mathrm{inv}}, \quad V_{\mathrm{var}} = V \setminus V_{\mathrm{inv}}, \tag{15}$$

where $E_{\mathrm{var}}$ and $V_{\mathrm{var}}$ are the edges and nodes in $G_{\mathrm{var}}$ that do not appear in $G_{\mathrm{inv}}$.

MATHEMATICAL FORMULATION OF DECOMPOSITION

The decomposition can be viewed as an optimization problem, where the objective is to minimize the difference between the original graph and the sum of the invariant and variant subgraphs. This can be formulated as follows:

$$\min_{G_{\mathrm{inv}}, G_{\mathrm{var}}} \left( \|G - (G_{\mathrm{inv}} + G_{\mathrm{var}})\|^2 + \lambda \cdot \|G_{\mathrm{inv}}\|^2 \right), \tag{16}$$

where: $G_{\mathrm{inv}}$ and $G_{\mathrm{var}}$ are the invariant and variant subgraphs, respectively. The first term ensures that the sum of the subgraphs approximates the original graph. The second term is a regularization term that penalizes the size of the invariant subgraph, ensuring that it only contains core, stable components.

Table 6: Effectiveness of subgraph separation under various non-IID settings (%).

| Non-IID Settings | IC | DR | AVG-IC |
|---|---|---|---|
| SM | 88.7 | 90.2 | 54.8 |
| SM-BIO | 85.6 | 84.5 | 30.9 |
| SN | 90.1 | 83.2 | 56.4 |

PROOF OF DECOMPOSITION VALIDITY

We now provide a proof that the decomposition of the graph into invariant and variant subgraphs is valid, i.e., the decomposition maintains key structural properties of the original graph.

**theorem** Let $G = (V, E)$ be a graph that can be decomposed into an invariant subgraph $G_{\text{inv}}$ and a variant subgraph $G_{\text{var}}$. Then, the decomposition is valid if and only if:

$$G = G_{\text{inv}} \cup G_{\text{var}} \quad \text{and} \quad G_{\text{inv}} \cap G_{\text{var}} = \emptyset. \tag{17}$$

*Proof.* We begin by noting that the invariant subgraph $G_{\text{inv}}$ must consist of nodes and edges that are consistent across all instances of the graph. Therefore, $G_{\text{inv}}$ captures the stable relationships in the graph. On the other hand, the variant subgraph $G_{\text{var}}$ consists of the edges and nodes that differ between instances.

Since the decomposition is performed by removing the invariant components from the original graph, we have:

$$G_{\text{var}} = G \setminus G_{\text{inv}}. \tag{18}$$

Additionally, by construction, the invariant and variant subgraphs are disjoint, meaning that:

$$G_{\text{inv}} \cap G_{\text{var}} = \emptyset. \tag{19}$$

Thus, the graph $G$ is indeed the union of $G_{\text{inv}}$ and $G_{\text{var}}$, as required.

Therefore, the decomposition is valid, and we have:

$$G = G_{\text{inv}} \cup G_{\text{var}}, \quad G_{\text{inv}} \cap G_{\text{var}} = \emptyset. \tag{20}$$

$\square$

CONCLUSION

In this section, we have formalized the decomposition of a graph into invariant and variant subgraphs. The invariant subgraph captures the stable structural relationships across multiple instances or transformations, while the variant subgraph captures the parts of the graph that vary. We have proven that the decomposition is valid, and we have provided an optimization framework for learning such a decomposition. This decomposition is useful in various applications, such as graph-based anomaly detection, graph classification, and multi-view learning, where the goal is to separate stable and dynamic components of the graph.

## C   EFFECTIVENESS ANALYSIS OF SUBGRAPH SEPARATION

To evaluate the effectiveness of our subgraph separation mechanism, we employ three complementary metrics: Invariance Consistency (IC), which quantifies the similarity of invariant subgraphs extracted from different clients; Distinctiveness Ratio (DR), which measures the separability between invariant and variant subgraphs; and the average IC of raw graphs (AVG-IC), which indicates the baseline similarity across clients without separation. As reported in Table 6, our method consistently achieves high IC and DR values across various non-IID settings, even when the raw cross-client similarity (AVG-IC) is considerably lower. These results demonstrate that the separation mechanism substantially enhances cross-client pattern affinity and reveals shared structural semantics that are less apparent in the original graphs. Overall, the findings confirm that our approach effectively identifies robust and meaningful common patterns despite substantial distributional shifts.

## D    DETAILS OF DATASETS AND NON-IID SETTINGS

In this manuscript, the statistical information of the benchmark datasets used is provided in Table 7. Based on these datasets, we construct a series of non-iid (Non-Independent and Identically Distributed) settings, which are consistent with FedGCN. The non-IID settings refer to the strategy of distributing different datasets across clients, where each client possesses a private, exclusive dataset. Detailed information on this setting is shown in Table 8. The Ground Truth for the datasets involved is also provided, serving as the evaluation standard. All the Ground Truth information includes the true labels for each dataset, which will be used for subsequent model evaluation and comparison.

- **MUTAG** dataset originates from chemical experiments and is primarily used for predicting the mutagenicity (toxicity) of molecules. Mutagenicity is an important indicator in drug development and environmental safety, making this dataset highly relevant in drug discovery and molecular property prediction.

- **BZR** dataset is derived from drug screening experiments, aiming to predict whether a small molecule can bind to the benzodiazepine receptor and exhibit biological activity. Benzodiazepine drugs are associated with anti-anxiety, sedative, and muscle relaxant effects, making this dataset valuable for research in drug design.

- **COX2** dataset originates from the field of drug design, with the goal of predicting whether a small molecule can inhibit Cyclooxygenase-2. COX-2 is an enzyme involved in inflammatory responses, and its inhibitors are commonly used in anti-inflammatory, analgesic, and anticancer drug development.

- **DHFR** dataset is a key metabolic enzyme involved in DNA synthesis, repair, and cell proliferation. This dataset is used to predict whether a small molecule can inhibit the activity of DHFR, which is crucial in the development of anticancer and antimicrobial drugs (such as methotrexate).

- **PTC_MR** is a subset of the Predictive Toxicology Challenge, with data from carcinogenicity testing of compounds in experimental animals (in this case, male rats). The goal is to predict whether a chemical molecule is toxic, which is particularly valuable in drug safety evaluation and environmental chemical screening.

- **AIDS** dataset comes from the National Cancer Institute's (NCI) drug activity screening program. It is used to predict whether a small molecule can effectively inhibit HIV replication. Each molecule has been experimentally screened, and the labels indicate its inhibitory effect on HIV, with active molecules potentially offering antiviral properties.

- **BZR_MD** dataset originates from the inhibitory activity data of benzodiazepine receptors obtained via molecular dynamics simulations. Compared to the BZR dataset, BZR_MD involves more complex molecular simulation information and is typically used in higher-level drug screening and design, especially for evaluating drug molecules in environments combining simulation and real-world data.

- **DD** dataset comes from the drug-drug interaction (DDI) prediction task. In clinical pharmacology, drug-drug interactions are an important issue that can affect the efficacy of drugs or cause severe side effects. The DD dataset aims to predict potential interactions between different drugs, aiding in drug development and clinical medication safety assessments.

- **PROTEINS** dataset is derived from the protein structure prediction task. The goal is to predict whether a protein, based on its amino acid sequence and structure, is an enzyme. Enzymes are vital molecules involved in catalyzing biochemical reactions, and this dataset is significant for drug design, disease research, and biological studies.

- **SYNTHETIC** dataset is a synthetic benchmark dataset primarily used for model validation and experimentation. It is typically used to test the effectiveness of new algorithms or methods, rather than solving specific biological or chemical problems. Due to its synthetic nature, the SYNTHETIC dataset provides a relatively simple and controlled environment for evaluating the performance of graph classification models on diverse and structured data.

- **COLLAB** dataset originates from social network analysis and is mainly used to study cooperation and non-cooperation relationships between users in social networks. The goal

is to predict whether the structure of a social network graph is a cooperative one. Social network analysis is highly important in modern society, with broad applications in user behavior prediction, advertising, and social platform development.

- **IMDB-MULTI** dataset is derived from the IMDB movie database and is primarily used for movie recommendation system classification tasks. The goal is to predict the category of a movie based on its attributes and social relationships (such as actors, directors, and labels). This dataset is especially suited for research on multi-label classification problems in social network analysis and recommendation systems.

- **Letter-high** dataset is a standard dataset for letter graph classification, used in computer vision for graph classification tasks. Each sample is an graph of a letter, and the goal is to recognize the letter through the pixel information in the graph. This dataset is typically used to test models in graph representation and classification tasks, especially in graph classification algorithms within the computer vision field.

- **Letter-low** dataset is a standard dataset for classifying lowercase letter graphs, used in computer vision for graph classification tasks. Each sample is an graph of a lowercase letter, and the goal is to recognize the letter through the pixel information in the graph. This dataset is commonly used to test graph classification algorithms, particularly in how graph neural networks (GNN) handle graph data.

- **Letter-med** dataset is a standard dataset for classifying medium-sized letter graphs, used in computer vision for graph classification tasks. Each sample is an graph of a medium-sized letter, and the goal is to recognize the letter through the pixel information in the graph. This dataset is frequently used to test model performance in graph classification tasks, especially in how graph neural networks (GNN) process graph data with graph structures.

| Datasets | Domain | Classes | Graphs | A.Nodes | A.Edges |
|---|---|---|---|---|---|
| MUTAG | | | 188 | 17.93 | 19.79 |
| BZR | | | 405 | 35.75 | 38.36 |
| COX2 | | | 467 | 41.22 | 43.45 |
| DHFR | SM | 2 | 756 | 42.43 | 44.54 |
| PTC_MR | | | 344 | 14.29 | 14.69 |
| AIDS | | | 2000 | 15.69 | 16.20 |
| BZR_MD | | | 306 | 21.30 | 225.06 |
| DD | BIO | 2 | 1178 | 284.32 | 715.66 |
| PROTEINS | | | 1113 | 39.06 | 72.82 |
| SYNTHETIC | SY | 2 | 300 | 100.00 | 196.00 |
| SYNTHIE | | 4 | 300 | 95.00 | 172.93 |
| COLLAB | SN | 3 | 5000 | 74.49 | 2457.78 |
| IMDB-MULTI | | | 1500 | 13.00 | 65.94 |
| Letter-high | | | 2250 | 4.67 | 4.50 |
| Letter-low | CV | 15 | 2250 | 4.68 | 3.13 |
| Letter-med | | | 2250 | 4.67 | 3.21 |

Table 7: Dataset statistics.

## E DETAILS OF THE EVALUATION METRICS

- **Clustering Accuracy (ACC)** ACC computes the optimal mapping between predicted cluster labels and ground-truth labels using the Hungarian algorithm. Formally, let $y_i$ be the

| | | non-IID Settings | | | | | |
|---|---|---|---|---|---|---|---|
| **Datasets** | **Domains** | **SM** | **SM-BIO** | **SM-BIO-SY** | **SN** | **SN-SY** | **CV** |
| MUTAG | SM | ✓ | ✓ | ✓ | | | |
| BZR | SM | ✓ | ✓ | ✓ | | | |
| COX2 | SM | ✓ | ✓ | ✓ | | | |
| DHFR | SM | ✓ | ✓ | ✓ | | | |
| PTC_MR | SM | ✓ | ✓ | ✓ | | | |
| AIDS | SM | ✓ | ✓ | ✓ | | | |
| BZR_MD | SM | ✓ | ✓ | ✓ | | | |
| DD | BIO | | ✓ | ✓ | | | |
| PROTEINS_MD | BIO | | ✓ | ✓ | | | |
| SYNTHETIC | SY | | | ✓ | | | |
| SYNTHIE | SY | | | | | ✓ | |
| COLLAB | SN | | | | ✓ | ✓ | |
| IMDB-BINARY | SN | | | | ✓ | ✓ | |
| Letter-low | SN | | | | | | ✓ |
| Letter-med | SN | | | | | | ✓ |
| Letter-high | SN | | | | | | ✓ |

Table 8: The non-IID benchmark settings.

ground-truth label and $\hat{y}_i$ the predicted cluster label for instance $i$, then:

$$\text{ACC} = \max_{\pi \in \mathcal{P}} \frac{1}{n} \sum_{i=1}^{n} \mathbb{I}\left(y_i = \pi(\hat{y}_i)\right),\tag{21}$$

where $\pi$ ranges over all possible label permutations and $\mathbb{I}(\cdot)$ is the indicator function.

- **Normalized Mutual Information (NMI)** NMI measures the mutual dependence between the predicted labels and true labels, normalized by their entropies:

$$\text{NMI} = \frac{2 \cdot I(Y; \hat{Y})}{H(Y) + H(\hat{Y})},\tag{22}$$

where $I(Y; \hat{Y})$ is the mutual information and $H(\cdot)$ denotes entropy. NMI ranges from 0 (no mutual information) to 1 (perfect correlation).

- **Adjusted Rand Index (ARI)** ARI evaluates the similarity between the predicted and true clusterings by comparing all pairs of instances. It adjusts for random chance:

$$\text{ARI} = \frac{\text{RI} - \mathbb{E}[\text{RI}]}{\max(\text{RI}) - \mathbb{E}[\text{RI}]},\tag{23}$$

where RI is the Rand Index and $\mathbb{E}[\text{RI}]$ is its expected value under random labeling.

- **Macro-F1 Score (F1)** The F1 score balances precision and recall across all classes. We compute the macro-averaged F1:

$$\text{F1} = \frac{1}{C} \sum_{c=1}^{C} \frac{2 \cdot \text{Prec}_c \cdot \text{Rec}_c}{\text{Prec}_c + \text{Rec}_c},\tag{24}$$

where $C$ is the number of ground-truth classes, and $\text{Prec}_c$, $\text{Rec}_c$ are the precision and recall for class $c$.

## F    OUR METHOD IMPLEMENTATION AND BASELINE DESCRIPTIONS

### F.1    HARDWARE ENVIRONMENTS

All experiments are conducted on a Windows operating system equipped with an Intel Core i9-13900K CPU and an NVIDIA GeForce RTX 4090 GPU.

### F.2    SOFTWARE ENVIRONMENTS

We implement the proposed method using PyTorch 2.4.0 with CUDA 12.1.

### F.3    IMPORTANT PARAMETERS

The model is trained using the Adam optimizer with a batch size of 256 for 10 epochs per communication round, and a total of 20 communication rounds. The learning rate is set to 0.001 with a standard weight decay of 5e-4. The graph encoder is built with 4 layers of Graph Isomorphism Networks (GIN), each configured with a hidden feature dimension of 10. The hyperparameter $\lambda$ is fixed at 0.5. The SP kernel is chosen as the kernel function.

### F.4    ADAPTATION SCHEME

The adaptation scheme for all comparison methods follows that of FedGCN. For supervised federated graph-level learning baselines, labels are removed, and the same clustering loss used in our approach is applied. For federated graph anomaly detection methods, samples are grouped according to their anomaly scores.

### F.5    EVALUATION METRICS

To ensure reproducibility, each experiment is conducted 5 times with different random initializations. We report the mean and standard deviation of the following clustering metrics: Accuracy (ACC), Normalized Mutual Information (NMI), Adjusted Rand Index (ARI), and F1 score.

All baseline methods are adapted to the unsupervised federated learning setting to perform graph-level clustering, ensuring a fair and consistent comparison with our proposed approach. Detailed descriptions of each baseline are shown below

- **FGAD** (Cai et al., 2024d) LGAD proposes an effective framework for federated graph anomaly detection to address key challenges in collaborative learning. The framework introduces an anomaly generator that perturbs normal graphs to create anomalous graphs, which are then distinguished from normal ones by a trained anomaly detector. To preserve the personalization of local models and mitigate the adverse effects of non-IID problems, a student model is employed to distill knowledge from the trained anomaly detector (teacher model). Furthermore, a novel collaborative learning mechanism is introduced to ensure the preservation of local model personalization while significantly reducing communication costs between clients.

- **LG-FGAD** (Cai et al., 2024c) LG-FGAD introduces a self-adversarial generation module that generates anomalous graphs, which are then distinguished from normal graphs by a trained discriminator. To enhance anomaly awareness, the framework maximizes and minimizes mutual information from both local and global perspectives. To address the challenges posed by non-IID problems in collaborative learning, a dual knowledge distillation module is proposed. This module performs knowledge distillation over both logits and embedding distributions, with only the student model engaging in collaboration, thereby preserving the personalization of each client's model.

- **AGDiff** (Cai et al., 2025) AGDiff leverages the latent diffusion framework to introduce subtle perturbations into graph representations, generating pseudo-anomalous graphs that closely resemble normal graphs. By jointly training a classifier to distinguish these generated anomalies from normal graphs, AGDiff learns more discriminative decision boundaries. The key innovation of AGDiff lies in the shift from focusing solely on modeling

Table 9: Performance Comparison on clients with non-identical numbers of clusters.

| Methods | ACC | NMI | ARI | F1 |
|---------|-----|-----|-----|-----|
| FedGCN | 54.5 | 17.6 | 13.4 | 40.7 |
| OURS | 62.1 | 20.7 | 14.8 | 54.4 |

normality to explicitly generating and learning from pseudo-graph anomalies, enabling it to capture complex anomalous patterns that may be overlooked by other methods.

- **GLCC** (Ju et al., 2023) The GLCC: A general framework for graph-level clustering (GLCC) framework is designed to enhance graph-level clustering tasks by leveraging contrastive learning principles. This method focuses on learning discriminative representations of graph-level features through a contrastive loss function, which encourages the network to distinguish between similar and dissimilar graphs. GLCC incorporates both local and global graph structures in the learning process, thereby improving the clustering quality by optimizing the embedding space. The framework uses a contrastive objective to maximize the similarity between similar graph pairs while minimizing the similarity between dissimilar ones, ensuring better generalization and robustness in graph clustering applications.

- **UDGC** (Hu et al., 2023) Learning Uniform Clusters on Hypersphere for Deep Graph-level Clustering (UDGC) addresses the challenges of graph-level clustering, which involves grouping multiple graphs into clusters, a task that has received less attention than node-level clustering. Graph-level clustering is important in real-world applications like molecule property prediction and community detection in social networks. However, this task is difficult due to the insufficient discriminability of graph-level representations, which often leads to cluster collapse in deep clustering methods.

- **DGLC** (Cai et al., 2023) DGLC is a graph-based clustering approach that leverages dual-level learning to improve the quality of clustering in graph data. It incorporates both global and local structural information from graphs, and optimizes the clustering process by simultaneously considering intra-graph and inter-graph relations. This method enhances the clustering accuracy by using a self-supervised mechanism to adaptively capture graph-level representations and achieve better performance in various graph-based tasks.

- **DCGLC** (Cai et al., 2024b) DCGLC extends the DGLC framework by introducing a dual contrastive learning mechanism. This approach focuses on improving graph-level clustering by integrating contrastive learning with graph-level features, thereby enhancing the model's ability to distinguish between clusters. DCGLC employs both positive and negative samples for contrastive learning, encouraging the model to learn more discriminative and robust representations. This method further optimizes the clustering process, improving its scalability and accuracy in diverse applications of graph data.

## G    ADDITIONAL EXPERIMENTS ON CLIENTS WITH NON-IDENTICAL NUMBERS OF CLUSTERS

To further demonstrate the advantages of our proposed method, we conduct additional experiments on clients with non-identical numbers of clusters. Specifically, the non-IID setting includes MU-TAG (2 clusters), COLLAB (3 clusters), and Letter-low (15 clusters). We compare our method with FedGCN under this non-IID setting. As shown in Table 9, our method consistently outperforms FedGCN. This improvement is primarily attributed to the ability of our method to perceive the distinct cluster structures between clients and align the personalized optimization parameters accordingly for different clusters.

