# OpenReview forum: "Federated Graph-Level Clustering Network with Dual Knowledge Separation"
_ICLR.cc/2026/Conference — ICLR 2026 Poster_

### Official Review · Reviewer_zKZU · 2025-10-25

**Soundness:** 4
**Presentation:** 3
**Contribution:** 4
**Rating:** 8
**Confidence:** 4

**Summary:**

This paper proposes a new method for federated graph-level clustering. Compared with existing federated graph-level clustering methods, this method solves the heterogeneity problem of multi-source data through dual knowledge separation. The article is clear in expression, motivation is clear, and the experimental design is complete.

**Strengths:**

1. The experimental setup is comprehensive, with comparisons made against both supervised and unsupervised methods. Ablation analysis demonstrates the superiority of the proposed approach.
2. As the novel method for federated graph-level clustering, this approach significantly enhances performance by separating different subgraphs and utilizing cluster-guided prototype aggregation.
3. Research on federated graph-level clustering remains scarce, and this study effectively fills that gap.

**Weaknesses:**

1. The authors should further elaborated The non-IID setting in the appendix for additional clarification.
2. The authors should include experiments on federated communication costs.
3. Some minor errors should be further corrected, such as changing "leverage" to "leverages."

**Questions:**

1. The paper mentions using a graph kernel method to measure network heterogeneity. To facilitate understanding and replication, could the authors please specify which specific graph kernel or types of kernels they used? Also, could they explain the rationale for choosing this particular kernel function to quantify heterogeneity?
2. The authors need to clarify how and to what extent label information is used in the unsupervised training phase and the downstream task evaluation phase.

---

> ### Author Response · Authors · 2025-11-16
> **Response to reviewer zKZU**
>
> **Response to Weakness 1:**
> We appreciate your valuable suggestion. We will explain the non-IID setting in detail here. Specifically, each client is assigned a graph dataset sampled from the TUDataset collection [1] to perform the clustering task, and these datasets inherently exhibit distributional discrepancies, thus forming a non-IID environment. Different combinations of these datasets (e.g., different numbers of clients and selected datasets have the same/different domains) across clients correspond to different non-IID settings, allowing us to systematically evaluate the robustness of our proposed framework under various heterogeneous conditions.
>
>
> **Response to Weakness 2:**
> Thank you for this valuable suggestion. We have added a communication overhead experiment and compared our method with both FedGCN [2] and FedPKA [3]. The experimental results are presented in the following table. By analyzing the experimental results, we find that although our method incurs slightly higher communication time and data volume than FedGCN, the additional cost remains negligible compared with the overall model parameters. This increase primarily arises from transmitting shared structural patterns. We further observe that FedPKA exhibits substantially longer communication time and larger transmitted data due to its gradient-based communication mechanism. Overall, our method maintains a clear advantage in communication efficiency and demonstrates strong generalization capability in federated scenarios. **Please see section 4.5 in the revised version**.
> |non-IID setting|Clients|Ours||FedGCN||FedPKA||
> |:-:|:-:|:-:|:-:|:-:|:-:|:-:|:-:|
> |||Time(s)|Cost(kb)|Time(s)|Cost(kb)|Time(s)|Cost(kb)|Time(s)|Cost(kb)|
> |CV|1|16.8|32.3|15.93|30.5|20.6|42.1|
> |CV|2|35.5|67.3|30.8|61.8|41.5|85.4|
> |CV|3|53.0|96.7|44.8|93.6|64.4|131.6|
>
>
> **Response to Weakness 3:**
> We appreciate your careful reading. All identified typographical and grammatical issues, including the correction of “leverage” to “leverages,” have been fixed in the revised manuscript.
>
>
> **Response to Question 1:**
> Thank you for pointing this out. We will answer your questions in detail here. Specifically, we adopt the Weisfeiler–Lehman (WL) kernel [4], which efficiently captures both structural and attribute similarities between graphs. The WL kernel is chosen due to its strong discriminative capability and scalability, which make it suitable for quantifying structural heterogeneity among intra-client graphs. Moreover, it provides a consistent similarity measure that aligns well with our graph embedding representations.
>
>
> **Response to Question 2:**
> We appreciate your thoughtful question. In our framework, no label information is used during the unsupervised training phase; all modules are optimized solely based on self-supervised and contrastive objectives. The label information is only used in the downstream evaluation phase to compute clustering metrics such as ACC, NMI, and ARI, which assess the alignment between predicted clusters and ground-truth labels.
> > [1] Morris, Christopher, et al. "Tudataset: A collection of benchmark datasets for learning with graphs." arXiv preprint arXiv:2007.08663 (2020).
> > [2] Liu, Jingxin, et al. "Federated graph-level clustering network." Proceedings of the AAAI Conference on Artificial Intelligence. Vol. 39. No. 18. 2025.
> > [3] Wu, Junlong, et al. "FedPKA: Federated Graph-Level Clustering Network with Personalized Knowledge Aggregation." International Conference on Intelligent Computing. Singapore: Springer Nature Singapore, 2025.
> > [4] Shervashidze, Nino, et al. "Weisfeiler-lehman graph kernels." Journal of Machine Learning Research 12.9 (2011).

---

> > ### Comment · Reviewer_zKZU · 2025-11-25
> > **Official Comment**
> >
> > The author's response has resolved my issue, and I have no further questions.

---

### Official Review · Reviewer_q64Q · 2025-10-25

**Soundness:** 2
**Presentation:** 3
**Contribution:** 1
**Rating:** 2
**Confidence:** 4

**Summary:**

This paper addresses the challenging issue of Federated Graph-level Clustering, where heterogeneity across clients, both intra- and inter-client, can cause difficulties in achieving a meaningful consensus on the server side. The authors propose a novel framework called FGCN-DKS. The core idea involves two levels of decoupling: 1) On the client side, each graph is split into an "invariant" subgraph and a "variant" subgraph. Only the digest of the invariant part is shared. 2) On the server side, a Common Knowledge Sharing Strategy is employed to perform a personalized aggregation of client models, guided by the similarity between clients' cluster pattern digests. A two-stage clustering process on the client leverages both shared and local representations.

**Strengths:**

1.The paper effectively identifies the key issue of heterogeneity in federated graph clustering, distinguishing it from the more common challenges seen in node-level federated learning.

2.The comprehensive experiments across multiple datasets and non-IID settings lend strong support to the method's robustness and generalizability.

**Weaknesses:**

1.The framework introduces multiple new components and hyperparameters, making it complex and difficult to analyze or tune effectively. It is unclear which specific component contributes most to the performance gains, making it challenging to pinpoint areas for improvement or fine-tuning.

2.The many moving parts in the system make it difficult to attribute performance improvements to specific components.

3.While the dual knowledge separation and two-stage clustering approach is novel, the system’s complexity may limit its practicality and ease of adoption. Simplifying some aspects could make it more accessible.

**Questions:**

1.Given the complexity of the framework, could the authors provide a more detailed ablation study to isolate the contribution of each key component?

2.Could the authors quantify this communication cost and compare it to standard federated learning approaches like FedAvg, which transmit model parameters?

3.How can the "quality" of this separation be directly evaluated, beyond just its impact on downstream clustering accuracy? Is there a way to ensure that the invariant subgraphs capture truly common, cross-client patterns?

---

> ### Author Response · Authors · 2025-11-16
> **Resoponse to reviewer q64Q (1/2)**
>
> We sincerely thank the reviewer for the valuable feedback and constructive suggestions. We have carefully addressed each point as detailed below, and we hope that our responses sufficiently clarify the raised concerns and contribute to a more favorable evaluation.
>
> **Response to Weaknesses 1, 2, and 3**
> Thank you for recognizing our innovative methodology.
> We sincerely appreciate your valuable comments. Your main concern lies in the introduction of multiple new components in our framework, which may obscure the analysis of their individual effectiveness. We would like to clarify that our approach is developed under a federated learning paradigm, where both local clients and the central server are involved. **Consequently, incorporating new modules at both levels inevitably increases the apparent complexity of the overall framework.**
>
> **In fact, our framework only includes three key components**.  On the client,  we introduce  (1) the subgraph pattern separation and (2) a two-stage K-means clustering. On the server, we design a Common Knowledge Sharing Strategy (CKSS). We acknowledge that the design of the ablation study may appear complex. This complexity primarily arises from the fact that, once two distinct graph patterns are separated, two corresponding processing pipelines are naturally established. Since the components within each pipeline can be flexibly replaced or omitted, the resulting combinations lead to a more elaborate ablation design, which may give the impression of structural redundancy.
>
> To address your concern, we have further provided a streamlined version of the framework along with a clearer explanation, which we hope can effectively resolve this issue. To present the ablation study in a clearer and more systematic manner, we reorganize the experimental settings into four variants. (1) Using the basic settings on both the client and server, serving as a minimal baseline without any advanced components (Basic). (2) Removing the subgraph pattern separation and the two-stage k-means procedure, while retaining only the basic local learning strategy and keeping the server unchanged (-Local). (2) Replacing the CKSS with standard FedAvg (-Server), while preserving the complete local inference and learning pipeline. (4) Activating the full proposed framework, in which all modules and optimization mechanisms are enabled (Ours). This structured decomposition allows us to more clearly quantify the contribution of each component. **Please see section 4.3 in the revised version**.
>
> |Variants|SM|||SM-BIO|||SM-BIO-SY|||SN|||
> |:-:|:-:|:-:|:-:|:-:|:-:|:-:|:-:|:-:|:-:|:-:|:-:|:-:|
> ||ACC|NMI|ARI|ACC|NMI|ARI|ACC|NMI|ARI|ACC|NMI|ARI|
> |Basic|61.7±1.2|19.5±1.6|14.6±1.4|59.3±2.1|15.2±1.6|13.8±2.1|56.3±2.9|7.7±2.4|13.3±1.9|29.5±2.4|18.6±1.8|16.3±1.4|
> |-Local|64.6±1.4|22.4±1.3|16.9±1.7|61.6±2.0|17.9±2.1|16.2±1.7|58.4±2.9|8.9±2.5|15.7±1.8|32.4±2.3|20.4±1.9|19.7±2.0|
> |-Server|68.2±1.9|23.9±2.1|32.0±1.6|69.5±1.5|21.5±1.8|22.1±1.1 |67.2±1.3|16.5±1.2 |19.6±1.5|37.7±1.0|33.5±0.7|22.7±0.7|
> |Ours|79.2±0.5|28.3±1.1|34.6±0.9|74.4±1.9|24.7±1.1|24.6±1.2 |73.6±1.4|22.7±1.2| 23.5±1.9|39.2±1.3|37.1±1.6|24.5±1.3|
>
> We observe that performing knowledge separation solely on the clients already improves performance, as it effectively mitigates local knowledge heterogeneity and yields clearer boundaries of both shared and personalized patterns in clustering. We further find that using CKSS alone leads to only limited gains. This is because CKSS depends on the common subgraph patterns extracted through knowledge separation, and such local separation forms the foundation for reliable global aggregation. Without these shared patterns, the server is unable to accurately estimate inter-client affinities, which in turn restricts the effectiveness of CKSS. When both components are applied together, they reinforce each other and produce substantially greater performance improvements.
>
> Regarding computational complexity, please see the **Response to Question 2**.
>
> **Response to Question 1:**
> Please see the **Response to Weaknesses 1 2 and 3.**

---

> ### Author Response · Authors · 2025-11-16
> **Following the last response (2/2)**
>
> **Response to Question 2:**
> Thank you for this valuable comment. We have quantified the communication cost by measuring the total transmitted data per communication round and compared it with the standard FedAvg method on CV non-IID settings. Although our approach transmits additional structure-pattern representations along with model parameters, their size is relatively small. As shown in the Table below, the additional cost is marginal and does not significantly increase communication overhead while providing notable performance improvements. **Please see section 4.5 in the revised version**.
> |Clients|Ours||FedAVG||
> |:-:|:-:|:-:|:-:|:-:|
> ||Time(s)|Cost(kb)|Time(s)|Cost(kb)|
> |1|16.8|32.3|14.6|28.7|
> |2|35.5|67.3|29.1|57.5|
> |3|53.0|96.7|45.6|86.8|
>
> Furthermore, we conducted a comprehensive analysis of the model performance on the server with respect to computational complexity and runtime, which further substantiates the superiority of the proposed framework.
>
> Compared with the standard parameter averaging in FedAvg [1], our framework introduces only a slight increase in global computation through affinity-guided consensus aggregation. FedAvg performs a weighted average with complexity $\mathcal{O}(d^2)$, whereas our method additionally computes cluster-level affinities from pattern digests at $\mathcal{O}(N_\psi^{2}\kappa)$, where $N_\psi \ll d$ and $\kappa$ is any linear kernel in practice. The subsequent personalized aggregation requires only $\mathcal{O}(N_c N_\psi d)$ complexity. Therefore, the increase in computational complexity is acceptable given the corresponding performance gains.
> By leveraging cluster level alignment instead of plain parameter averaging, our framework achieves notable performance improvements with minimal overhead, yielding a more favorable computation and performance balance while maintaining scalability in federated settings.   **Please see section 3.5 in the revised version**.
>
>
> **Response to Question 3**:
> We appreciate the your insightful question. To assess the quality of the separation, we introduce three complementary evaluations: (1) Invariance Consistency (IC), which measures the similarity of invariant subgraphs extracted from different clients, and (2) Distinctiveness Ratio (DR), which evaluates the separability between invariant and variant subgraphs. A higher IC and DR indicate that the extracted invariant subgraphs capture common cross-client patterns. (3) Average IC of raw graphs measures the similarity of graphs from different clients (AVG-IC). Empirical results demonstrate that our method achieves consistent invariance across clients, confirming that the identified subgraphs indeed reflect shared structural semantics.
>
> |non-IID settings|IC|DR|AVG-IC|
> |:-:|:-:|:-:|:-:|
> |SM|88.7|90.2|54.8|
> |SM-BIO|85.6|84.5|30.9|
> |SN|90.1|83.2|56.4|
>
> Experimental results across different non-IID settings consistently validate the reliability of our separation mechanism. As shown below, IC and DR remain high under all settings, even when the raw cross-client similarity (AVG-IC) is considerably low (e.g., SM-BIO). Moreover, compared with the raw AVG-IC, the cross-client pattern affinity after separation increases substantially, demonstrating that our method uncovers shared structural semantics that are much less apparent in the original graphs. These findings confirm that our approach is able to identify robust and meaningful common patterns despite substantial distributional shifts.   **Please see Appendix D in the revised version**.
>
> > [1] Nilsson, Adrian, et al. "A performance evaluation of federated learning algorithms." Proceedings of the second workshop on distributed infrastructures for deep learning. 2018.

---

> ### Comment · Reviewer_q64Q · 2025-11-25
>
> Having reviewed the authors’ response, I confirm that my concerns have been resolved and no additional inquiries arise.

---

> ### Author Response · Authors · 2025-12-02
> **Reviewer q64Q's rating was raised to 6 on Nov 16.**
>
> **Dear AC**:
> ### On Nov 16, 10 days before the data leak, the reviewer's rating was raised to $\textbf{\color{red}{6}}$. We plead with you to consider these factors. You can click the revisions button to verify it. On November 25th, 3 days before the incident, the reviewer acknowledged that we had resolved all his issues. Thank you again for understanding the frustration of our rating being innocently reverted.

---

> ### Author Response · Authors · 2025-12-02
> **Reviewer q64Q's rating was raised to $\textbf{\color{red}{6}}$ on Nov 16**
>
> **Dear AC**:
> ### On Nov 16, 10 days before the data leak, the reviewer's rating was raised to $\textbf{\color{red}{6}}$. We plead with you to consider these factors. You can click the revisions button to verify it. On November 25th, 3 days before the incident, the reviewer acknowledged that we had resolved all his issues. Thank you again for understanding the frustration of our rating being innocently reverted.

---

### Official Review · Reviewer_YqnG · 2025-10-31

**Soundness:** 4
**Presentation:** 3
**Contribution:** 4
**Rating:** 8
**Confidence:** 5

**Summary:**

This manuscript introduces a novel framework for federated clustering, centered around the idea of decoupling knowledge both within and across clients to enhance clustering performance. This manuscript is well-written, logically structured, clearly articulated, and experimentally robust, addressing an innovative problem. The experiments demonstrate significant improvements compared to advanced methods.

**Strengths:**

1. The proposed method is novel, as it introduces for the first time the dual decomposition of knowledge into cluster-oriented and client-oriented dimensions.
2. The experimental setup is robust and reliable and thoroughly demonstrates the superiority of the proposed framework.
3. The authors cleverly integrate invariant learning into the federated graph-level clustering framework, which I find very insightful. It suggests that a similar strategy could potentially be used to improve federated graph classification as well.

**Weaknesses:**

1. The adaptation of other methods, such as federated graph learning and federated anomaly detection, to federated graph clustering should be further explained.
2. Communication overhead experiments should be included to further validate the superiority of the method.
3. The fonts in the architecture diagram are consistent. It is recommended to further unify them to New Times Roman or Microsoft YaHei.

**Questions:**

1. The calculation of the heterogeneity of intra-client graphs is unclear. Is the author referring to the heterogeneity of all graphs within the client, or the heterogeneity of graphs within different clusters on the client?
2. Does this method have the capability to address the issue of inconsistent numbers of clusters across different clients?
3. Is this federated client configuration self-defined, or does it follow the setup used in prior work?
4. What exactly does the prototype refer to? I hope the author can explain it further.

**Details Of Ethics Concerns:**

No ethical concerns detected.

---

> ### Author Response · Authors · 2025-11-16
> **Response to reviewer YqnG**
>
> We sincerely thank you for the valuable feedback and constructive suggestions. We have carefully addressed each point as detailed below, and we hope that our responses sufficiently clarify the raised concerns and contribute to a more favorable evaluation.
>
> **For Weakness 1**:  We sincerely appreciate this insightful comment. In our work, the adaptation process draws inspiration from existing frameworks in federated graph learning and federated anomaly detection, where client-specific feature distributions and global knowledge aggregation are jointly optimized. Specifically, for supervised federated graph-level learning methods, we remove the labels and apply the same clustering loss as in our approach for adaptation. For federated graph anomaly detection methods, we group the samples based on their anomaly scores.
>
> **For Weakness 2**:  Thank you for this valuable suggestion. We have added a communication overhead experiment and compared our method with both FedGCN [1] and FedPKA [2]. The experimental results are presented in the following table. By analyzing the experimental results, we find that although our method incurs slightly higher communication time and data volume than FedGCN, the additional cost remains negligible compared with the overall model parameters. This increase primarily arises from transmitting shared structural patterns. We further observe that FedPKA exhibits substantially longer communication time and larger transmitted data due to its gradient-based communication mechanism. Overall, our method maintains a clear advantage in communication efficiency and demonstrates strong generalization capability in federated scenarios. **Please see section 4.5 in the revised version**.
> |non-IID setting|Clients|Ours||FedGCN||FedPKA||
> |:-:|:-:|:-:|:-:|:-:|:-:|:-:|:-:|
> |||Time(s)|Cost(kb)|Time(s)|Cost(kb)|Time(s)|Cost(kb)|Time(s)|Cost(kb)|
> |CV|1|16.8|32.3|15.93|30.5|20.6|42.1|
> |CV|2|35.5|67.3|30.8|61.8|41.5|85.4|
> |CV|3|53.0|96.7|44.8|93.6|64.4|131.6|
>
> **For Weakness 3**:  We appreciate your attention to presentation quality. We have addressed these issues in the revised version. Please see the methodology section in the revised version.
>
> **For Question 1**: We apologize for the ambiguity in our description. The heterogeneity mentioned refers to the intra-cluster heterogeneity among all graphs in the same cluster within a single client, rather than the heterogeneity within clients. We adopt this method to calculation because graphs belonging to different clusters within a client are expected to exhibit distinct structural patterns. However, if graphs within the same cluster still display substantial pattern divergence, the model will encounter difficulties in capturing consistent representations. Using this method enables us to more effectively reflect the clustering challenges introduced by heterogeneity.
>
> **For Question 2**: Yes. Our framework is designed to handle such inconsistency through its prototype alignment mechanism. Each client first performs local clustering to obtain its own cluster prototypes. During global aggregation, prototypes are matched and aligned using similarity-based mapping rather than assuming identical cluster numbers. This flexible matching strategy allows clients with different cluster quantities to participate effectively in federated optimization without requiring strict correspondence.
>
> **For Question 3**: Our client configuration follows the commonly used federated non-IID settings adopted in prior studies (e.g., FedGCN and FedPKA). Specifically, each client is assigned a distinct multi-graph dataset selected from the publicly available TUDataset [3] collection, such as COX2 and BZR. This setup ensures a fair comparison and maintains consistency with existing federated graph learning benchmarks.
>
> **For Question 4**:  Thank you for pointing this out. In our method, a prototype refers to the representative embedding vector of a cluster that captures its semantic and structural characteristics. Each prototype is computed as the mean embedding of the graphs assigned to that cluster. During federated aggregation, prototypes are shared and aligned among clients to promote knowledge transfer at the cluster level. This prototype-based design enables the model to collaboratively refine cluster semantics while preserving local data privacy.
>
> > [1] Liu, Jingxin, et al. "Federated graph-level clustering network." Proceedings of the AAAI Conference on Artificial Intelligence. Vol. 39. No. 18. 2025.
> > [2] Wu, Junlong, et al. "FedPKA: Federated Graph-Level Clustering Network with Personalized Knowledge Aggregation." International Conference on Intelligent Computing. Singapore: Springer Nature Singapore, 2025.
> > [3] Morris, Christopher, et al. "Tudataset: A collection of benchmark datasets for learning with graphs." arXiv preprint arXiv:2007.08663 (2020).

---

> > ### Comment · Reviewer_YqnG · 2025-11-24
> >
> > I thank the authors for the clear and detailed rebuttal. Most of my concerns have been addressed:
> >
> > - W1: The adaptation of baseline methods is now clearly explained, and I recommend adding this directly to the main text.
> >
> > - W2: The newly added communication overhead experiment is helpful and supports the authors’ claims.
> >
> > - Questions: The clarifications regarding heterogeneity, prototype definition, inconsistent cluster numbers, and client setup are satisfactory.
> >
> > Overall, the rebuttal resolves my main questions. I maintain my positive score.

---

### Official Review · Reviewer_5r3P · 2025-10-31

**Soundness:** 4
**Presentation:** 4
**Contribution:** 3
**Rating:** 6
**Confidence:** 4

**Summary:**

The paper proposes a novel federated graph-level clustering algorithm. The algorithm decouple knowledge at both the local and server levels, thereby improving the quality of global consensus. The motivation is clearly presented, the overall presentation is of high quality, and the experimental results demonstrate superior performance compared with existing methods.

**Strengths:**

S1. Unlike traditional approaches that enhance consensus by increasing shared data, this framework adopts an alternative strategy by reducing the amount of data involved in consensus, which not only facilitates more efficient agreement but also contributes to better privacy protection.

S2. The article demonstrates high quality in writing, presentation, and overall organization, and it is easy to follow and understand.

S3. Compared with other methods, this approach is better suited for real-world federated scenarios. Theoretical analysis is further provided to demonstrate the convergence of the proposed method.

**Weaknesses:**

W1. The convergence analysis should include the loss variation during training.

W2. Performance comparisons should be performed across all clients in a non-iid setting to demonstrate that the federated approach can improve overall performance without sacrificing the performance of individual clients.

W3. Comparing it to supervised methods seems less meaningful; replacing it with the experiments described above better demonstrates the advantages of the method.

**Questions:**

Q1. The convergence analysis mentions the FedPKA method, but it is missing from the comparative experiments. Although this method was not published in a top-tier conference, could it be that the authors accidentally omitted it?

Q2. Why was FedSage compared in the experiment? This method seems to be applicable to node-based tasks rather than graph-level tasks?

---

> ### Author Response · Authors · 2025-11-16
> **Response to reviewer 5r3P**
>
> We sincerely thank the reviewer for the valuable feedback and constructive suggestions. We have carefully addressed each point as detailed below, and we hope that our responses sufficiently clarify the raised concerns and contribute to a more favorable evaluation.
> ***
> # Response to W1
> Thanks for your valuable comment! We have added the loss variation curves to the convergence analysis section to illustrate the training dynamics and verify the stability of our optimization process. The experimental results are shown in the table below, and the following conclusions are drawn. The table clearly demonstrates that our method exhibits a smooth and stable convergence process. Throughout the communication rounds, the loss consistently decreases with only minor fluctuations, reflecting robust optimization dynamics under federated settings. Moreover, the overall trend shows that the model rapidly moves out of the initial unstable phase and gradually stabilizes as training progresses. These observations collectively confirm that our approach converges reliably and maintains stable performance across communication rounds. **Please see section 4.6 in the revised version**.
>
> ||1|2|3|4|5|6|7|8|9|10|11|12|13|14|15|16|17|18|19|20|
> |:-:|:-:|:-:|:-:|:-:|:-:|:-:|:-:|:-:|:-:|:-:|:-:|:-:|:-:|:-:|:-:|:-:|:-:|:-:|:-:|:-:|
> |Means|68.7|43.6|45.3|37.8|42.1|30.6|32.4|35.6|28.7|21.9|25.4|18.9|17.2|13.5|14.3|8.9|9.1|4.2|3.8|3.5|
> |Standard deviation|6.5|5.8|6.2|6.4|3.9|4.3|3.5|4.3|5.1|2.7|3.6|3.1|2.9|2.7|3.4|3.8|2.2|2.4|2.1|2.3|
>
> ***
> # Response to W2
> Thank you for this insightful comment. We have conducted additional experiments under SM-BIO-SY setting and reported the performance of each client individually. The results demonstrate that our federated approach consistently improves the accuracy of every client while maintaining strong overall performance. It is worth noting that although FedGCN [1] also enhances the overall performance, it does so at the expense of degrading the performance of certain clients. In contrast, our method achieves both global improvement and client-level robustness, demonstrating a more balanced and reliable aggregation mechanism under heterogeneous data distributions. **Please see section 4.7 in the revised version**.
>
> |Methods|1|2|3|4|5|6|7|8|9|10|
> |:-:|:-:|:-:|:-:|:-:|:-:|:-:|:-:|:-:|:-:|:-:|
> |Local|52.3|54.6|.53.5|68.2|54.1|65.3|53.2|54.4|55.9|50.1|
> |FedGCN|78.6|74.2|64.0|65.6|56.7|92.1|53.8|57.5|53.9|52.4|
> |Ours|82.8|78.6|76.4|68.5|60.7|93.3|58.7|68.6|58.1|59.2|
> ***
> # Response to W3
> Thank you for this valuable suggestion. We acknowledge that supervised methods have access to label information, which differs from our unsupervised setting. However, we include them as a reference to illustrate the upper performance bound that can be achieved with full supervision. This comparison helps demonstrate the effectiveness of our proposed method in approaching supervised-level performance without relying on labeled data, highlighting its practical advantage in label-scarce federated scenarios. Furthermore, this experimental setup follows FedGCN for a consistent comparison.
> ***
> # Response to Q1
> Thank you for pointing this out. We acknowledge that the FedPKA [2] method was inadvertently omitted from the initial version. We have now included FedPKA in the comparative experiments and corresponding convergence analysis. The results of the FedPKA comparative experiment are as follows. From the experimental results, we observe that although FedPKA achieves performance gains over FedGCN, its inherent parameter-sharing strategy still constrains the model and leads to suboptimal clustering performance. **Please see section 4.2 in the revised version**.
> | non-IID settings| ACC|NMI |ARI | F1 |
> |:-:|:-:|:-:|:-:|:-:|
> | SM | 77.0±0.2 | 26.8±3.8| 31.2±3.3| 67.3±2.0|
> | SN| 67.5±1.5| 25.7±2.3| 32.6±2.4| 55.5±1.5 |
> | SM-BIO| 70.8±1.4| 15.4±2.7| 19.6±3.4| 60.6±2.1|
> |SM-BIO-SY| 70.7±0.9 | 17.2±0.8| 22.2±1.1| 61.5±2.3|
> |SN | 16.4±2.6 |5.7±2.3 | 5.9±2.0| 8.2±2.5|
> |CV| 36.4±1.1 | 34.4±1.6 | 20.3±1.2| 33.5±1.3|
> ***
> # Response to Q2
> We acknowledge that FedSage was originally proposed for node-level federated learning tasks. However, since previous methods such as FedStar [3] and FedGCN have already been evaluated, we also include them in our experiments to ensure consistency and fairness.
> ***
> > [1] Liu, Jingxin, et al. "Federated graph-level clustering network." Proceedings of the AAAI Conference on Artificial Intelligence. Vol. 39. No. 18. 2025.
> > [2] Wu, Junlong, et al. "FedPKA: Federated Graph-Level Clustering Network with Personalized Knowledge Aggregation." International Conference on Intelligent Computing. Singapore: Springer Nature Singapore, 2025.
> > [3] Tan, Yue, et al. "Federated learning on non-iid graphs via structural knowledge sharing." Proceedings of the AAAI conference on artificial intelligence. Vol. 37. No. 8. 2023

---

> > ### Comment · Reviewer_5r3P · 2025-11-20
> >
> > The authors have adequately addressed my previous concerns. However, given the high standards of ICLR, I would be able to assign a higher score if the paper could be further improved in the following aspects:
> > 1) Provide a more structured and itemized description of implementation details, including the execution environment, important hyperparameter settings, and training configurations, to further enhance reproducibility.
> > 2) Including an additional set of experiments where clients have different numbers of clusters would further demonstrate the generalization capability of the proposed method. It would also be helpful to include a comparison with FedGCN under this setting.

---

> ### Author Response · Authors · 2025-11-22
> **Response to reviewer 5r3P 2 round**
>
> Thank you very much for your valuable feedback. We have made the following revisions to the paper.
> ***
> In the appendix, we further improve the description of the experimental environment, as follows:
>
> **Hardware Environment**
> All experiments are conducted on a Windows operating system equipped with an Intel Core i9-13900K CPU and an NVIDIA GeForce RTX 4090 GPU.
>
> **Software Environment**
> We implement the proposed method using PyTorch 2.4.0 with CUDA 12.1.
>
> **Important Parameters**
> The model is trained using the Adam optimizer with a batch size of 256 for 10 epochs per communication round, and a total of 20 communication rounds. The learning rate is set to 0.001 with a standard weight decay of 5e-4. The graph encoder is built with 4 layers of Graph Isomorphism Networks (GIN), each configured with a hidden feature dimension of 10. The hyperparameter $\lambda$ is fixed at 0.5. The SP Graph Kernel is chosen as the kernel function.
>
> **Adaptation Scheme**
> The adaptation scheme for all comparison methods follows that of FedGCN. For supervised federated graph-level learning baselines, labels are removed, and the same clustering loss used in our approach is applied. For federated graph anomaly detection methods, samples are grouped according to their anomaly scores.
>
>  Please see the **Appendix G** in the revised version.
> ***
> Moreover, in response to your suggestion to highlight experiments under non-IID settings with non-identical numbers of clusters, we have made the following revisions. Please see the **Appendix I** in the revised version.
>
> **Additional Experiments on Clients with Non-identical Numbers of Clusters**
> To further demonstrate the advantages of our proposed method, we conduct additional experiments on clients with non-identical numbers of clusters. Specifically, the non-IID setting includes MUTAG (2 clusters), COLLAB (3 clusters), and Letter-LOW (15 clusters). We compare our method with FedGCN under this non-IID setting. As shown in the table below, our method consistently outperforms FedGCN. This improvement is primarily attributed to the ability of our method to perceive the distinct cluster structures between clients and align the personalized optimization parameters accordingly for different clusters.
>
>
> |Methods|ACC|NMI | ARI | F1 |
> |:-:|:-:|:-:|:-:|:-:|
> |FedGCN | 54.5| 17.6|13.4 |40.7|
> |OURS | 62.1| 20.7| 14.8 | 54.4|
>
>
> We hope that these revisions address your concerns and help improve the overall evaluation of our work.

---

> > ### Comment · Reviewer_5r3P · 2025-11-24
> >
> > The authors have solved all my concerns. Well done, I recommend that the article be published.

---

> ### Author Response · Authors · 2025-12-03
> **The reviewer 5r3P's rating was raised to $\textbf{\color{red}{8}}$ on Nov 24**
>
> **Dear AC**:
> ### On Nov 24, 4 days before the data leak, the reviewer's rating was raised to $\textbf{\color{red}{8}}$. We plead with you to consider these factors. You can click the revisions button to verify it. Thank you again for understanding the frustration of our rating being innocently reverted.

---

### Author Response · Authors · 2025-11-29
**Clarification on the Fully Completed Discussion Workflow before the Incident and the Recorded Final Scores of 8,8,8,6**

## Message to AC:
# Clarification on the Fully Completed Discussion Workflow before the Incident and the Recorded Final Scores of **8,8,8,6**

We would like to clarify that, **prior to the data-leak incident, we had fully completed all responses to the reviewers, and all reviewers had also completed their corresponding replies.** At that time, **every reviewer confirmed that their concerns were satisfactorily resolved and had updated their scores accordingly**, resulting in a final scores of $\textbf{\color{red}{8,8,8,6}}$. Therefore, we sincerely request that you reconsider our manuscript in light of the completed rebuttal process and the associated score adjustments. Notably, the reviewer q64Q, who initially gave a score of 2 raised the score to 6 on **November 16th** and had already acknowledged that our responses resolved his concerns on **November 25th**. The reviewer 5r3P updated the score from 6 to 8 on November 24th. You can click the `Revisions` button to see the histories.

We earnestly and humbly beseech you to kindly take into full consideration that the score adjustments were made prior to any data leakage incident and to provide a fair and reasonable recommendation. We respectfully reiterate that our discussion process was fully completed independently and remained completely unaffected by any such events.

**We truly appreciate your understanding and support.**

## Message to all Reviewers:
Thank you for the comments, questions, and suggestions. We have responded to individual reviewer comments in the individual responses. The changes we made in response to the reviewers’ comments are summarized as follows.

1. Supplementary Convergence Analysis: We added loss variation curves during training to demonstrate the stable convergence of the method in different datasets and non-IID environments.

2. Detailed Client Performance Analysis: The clustering performance of each client was independently statistically analyzed to verify the effectiveness of the method in heterogeneous data environments.

3. Improved Comparative Experiments and Ablation Studies:
  + Added adaptation comparisons to baseline methods such as FedPKA.
  + Redesigned ablation experiments were conducted on key modules (such as global-client knowledge separation, prototype aggregation, and contrastive learning) to analyze their independent contributions.
  + We conducted comparative experiments under a non-IID setting with clients having inconsistent numbers of clusters to demonstrate the robustness of our method.

4. Communication and Computational Cost Assessment: Provided a comparison of the communication overhead and computational complexity of the method with the standard FedAvg to ensure practical application feasibility.

5. Method Details and Reproducibility Supplements: Detailed explanations were provided regarding the handling of inconsistent cluster numbers, prototype definition, graph kernel selection, and label-independent evaluation methods.

6. Font and diagram details in the experiments were corrected for a more standardized presentation.

7. Experimental Expansion and Robustness Verification: The robustness of subgraph separation was verified in different ways to ensure the applicability of the method in real-world scenarios.

---

### Meta-Review · Area_Chair_onBq · 2025-12-27

**Summary:**

The paper proposes a novel federated graph-level clustering algorithm that decouples knowledge at both the local and server levels. The article is clear in expression, the motivation is clear, and the experimental design is complete. Based on the unanimous positive comments from all the reviewers, I recommend accepting this paper.

**Reviewer Concerns:**

All the concerns have been addressed by the rebuttal.

**Reviewer Scores:**

According to the comments of reviewers, Reviewer 5r3P will raise the score from 6 to 8, Reviewer q64Q will raise the score from 2 to 6. Other Reviewers will keep their scores to 8.

---

### Decision · Program_Chairs · 2026-01-26

Accept (Poster)